# Finding Bipartite Components in Hypergraphs

**Peter Macgregor**
School of Informatics
University of Edinburgh
`peter.macgregor@ed.ac.uk`

**He Sun**
School of Informatics
University of Edinburgh
`h.sun@ed.ac.uk`

## Abstract

Hypergraphs are important objects to model ternary or higher-order relations of objects, and have a number of applications in analysing many complex datasets occurring in practice. In this work we study a new heat diffusion process in hypergraphs, and employ this process to design a polynomial-time algorithm that approximately finds bipartite components in a hypergraph. We theoretically prove the performance of our proposed algorithm, and compare it against the previous state-of-the-art through extensive experimental analysis on both synthetic and real-world datasets. We find that our new algorithm consistently and significantly outperforms the previous state-of-the-art across a wide range of hypergraphs.

## 1   Introduction

Spectral methods study the efficient matrix representation of graphs and datasets, and apply the algebraic properties of these matrices to design efficient algorithms. Over the last three decades, spectral methods have become one of the most powerful techniques in machine learning, and have had comprehensive applications in a wide range of domains, including clustering [24, 31], image and video segmentation [26], and network analysis [25], among many others. While the success of this line of research is based on our rich understanding of Laplacian operators of graphs, there has been a sequence of very recent work studying *non-linear* Laplacian operators for more complex objects (i.e., hypergraphs) and employing these non-linear operators to design hypergraph algorithms with better performance.

### 1.1   Our contribution

In this work, we study the non-linear Laplacian-type operators for hypergraphs, and employ such an operator to design a polynomial-time algorithm for finding bipartite components in hypergraphs. The main contribution of our work is as follows:

First of all, we introduce and study a non-linear Laplacian-type operator $J_H$ for any hypergraph $H$. While we'll formally define the operator $J_H$ in Section 3, one can informally think about $J_H$ as a variant of the standard non-linear hypergraph Laplacian $L_H$ studied in [5, 20, 27], and this variation is needed to study the other end of the spectrum of $L_H$. We present a polynomial-time algorithm that finds some eigenvalue $\lambda$ and its associated eigenvector of $J_H$, and our algorithm is based on the following heat diffusion process: starting from an arbitrary vector $f_0 \in \mathbb{R}^n$ that describes the initial heat distribution among the vertices, we use $f_0$ to construct some 2-graph[1] $G_0$, and use the diffusion process in $G_0$ to represent the one in the original hypergraph $H$ and update $f_t$; this process continues until the time at which $G_0$ cannot be used to appropriately simulate the diffusion process in $H$ any more. At this point, we use the currently maintained $f_t$ to construct another 2-graph $G_t$

---

[1]Throughout the paper, we refer to non-hyper graphs as 2-graphs. Similarly, we always use $L_H$ to refer to the *non-linear* hypergraph Laplacian operator, and use $L_G$ as the standard 2-graph Laplacian.

35th Conference on Neural Information Processing Systems (NeurIPS 2021).

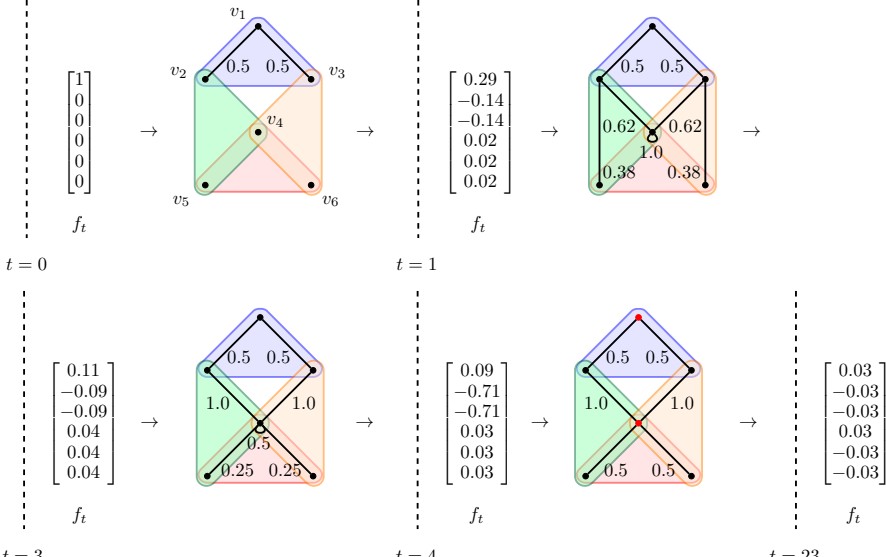

Figure 1: Illustration of our proposed diffusion process. In each time step, we construct a 2-graph $G$ based on the current vector $f_t$, and update $f_t$ with the $J_G$ operator. Notice that the graph $G$ changes throughout the execution of the algorithm, and that the final $f_t$ vector can be used to partition the vertices of $H$ into two well-connected sets (all the edges are adjacent to both sets), by splitting according to positive and negative entries. This specific example with $f_t$ values is generated by the publicly available implementation of our algorithm.

to simulate the diffusion process in $H$, and update $f_t$. This process continues until the vector $f_t$ converges; see Figure 1 for illustration. We theoretically prove that this heat diffusion process is unique, well-defined, and our maintained vector $f_t$ converges to some eigenvector of $J_H$. While this result is quite interesting on its own and forms the basis of our second result, our analysis shows that, for certain hypergraphs $H$, both the operator $J_H$ and $L_H$ could have $\omega(1)$ eigenvectors. This result answers an open question in [5], which asks whether $L_H$ could have more than 2 eigenvectors[2].

Secondly, we present a polynomial-time algorithm that, given a hypergraph $H = (V_H, E_H, w)$ as input, finds disjoint subsets $L, R \subset V_H$ that are highly connected with each other. The key to our algorithm is a Cheeger-type inequality for hypergraphs that relates the spectrum of $J_H$ and the bipartiteness ratio of $H$, an analog of $\beta_G$ studied in [28] for 2-graphs. Both the design and analysis of our algorithm is inspired by [28], however our analysis is much more involved because of the non-linear operator $J_H$ and hyperedges of different ranks. Our second result alone answers an open question posed by [33], which asks whether there is a hypergraph operator which satisfies a Cheeger-type inequality for bipartiteness.

The significance of our work is further demonstrated by extensive experimental studies of our algorithms on both synthetic and real-world datasets. In particular, on the well-known Penn Treebank corpus that contains $49,208$ sentences and over 1 million words, our *purely unsupervised* algorithm is able to identify a significant fraction of verbs from non-verbs in its two output clusters. Hence, we believe that our work could potentially have many applications in unsupervised learning for hypergraphs. Using the publicly available code of our implementation, we welcome the reader to explore further applications of our work in even more diverse datasets.

## 1.2 Related work

The spectral theory of hypergraphs using non-linear operators is introduced in [5] and generalised in [33]. The operator they describe is applied for hypergraph clustering applications in [20, 27]. There are many approaches for finding clusters in hypergraphs by constructing a 2-graph which approximates the hypergraph and using a 2-graph clustering algorithm directly [7, 19, 35]. Another

---

[2]We underline that, while the operator $L_G$ of a 2-graph $G$ has $n$ eigenvalues, the number of eigenvalues of $L_H$ is unknown because of its non-linearity. As answering this open question isn't the main point of our work, we refer the reader to Appendix C for detailed discussion.

approach for hypergraph clustering is based on tensor spectral decomposition [15, 18]. [21, 23, 36] consider the problem of finding densely connected clusters in 2-graphs. Heat diffusion processes are used for clustering 2-graphs in [10, 17]. [14] studies a different, flow-based diffusion process for finding clusters in 2-graphs, and [13] generalises this to hypergraphs. We note that all of these methods solve a different problem to ours, and cannot be compared directly. Our algorithm is related to the hypergraph max cut problem, and the state-of-the-art approximation algorithm is given by [34]. [28] introduces graph bipartiteness and gives an approximation algorithm for the 2-graph max cut problem. To the best of our knowledge, we are the first to generalise this notion of bipartiteness to hypergraphs. Finally, we note that there have been recent improvements in the time complexity for solving linear programs [11, 29] although we do not take these into account in our analysis since the goal of this paper is not to obtain the fastest algorithm possible.

## 2 Notation

**2-graphs.** Throughout the paper, we call a non-hyper graph a 2-graph [4, 8]. We always use $G = (V_G, E_G, w)$ to express a 2-graph, in which every edge $e \in E_G$ consists of two vertices in $V_G$ and we let $n = |V_G|$. The degree of any vertex $u \in V_G$ is defined by $d_G(u) \triangleq \sum_{v \in V_G} w(u, v)$, and for any $S \subseteq V$ the volume of $S$ is defined by $\mathrm{vol}_G(S) \triangleq \sum_{u \in S} d_G(u)$. Following [28], the *bipartiteness ratio* of any disjoint sets $L, R \subset V_G$ is defined by

$$\beta_G(L, R) \triangleq \frac{2w(L, L) + 2w(R, R) + w(L \cup R, \overline{L \cup R})}{\mathrm{vol}_G(L \cup R)}$$

where $w(A, B) = \sum_{(u,v) \in A \times B} w(u, v)$, and we further define $\beta_G \triangleq \min_{S \subset V} \beta_G(S, V \setminus S)$. Notice that a low $\beta_G$-value means that there is a dense cut between $L$ and $R$, and there is a sparse cut between $L \cup R$ and $V \setminus (L \cup R)$. In particular, $\beta_G = 0$ implies that $(L, R)$ forms a bipartite component of $G$. We use $D_G$ to denote the $n \times n$ diagonal matrix whose entries are $(D_G)_{uu} = d_G(u)$, for all $u \in V$. Moreover, we use $A_G$ to denote the $n \times n$ adjacency matrix whose entries are $(A_G)_{uv} = w(u, v)$, for all $u, v \in V$. The Laplacian matrix is defined by $L_G \triangleq D_G - A_G$. In addition, we define $J_G \triangleq D_G + A_G$, and $\mathcal{J}_G \triangleq D_G^{-1/2} J_G D_G^{-1/2}$. For any real and symmetric matrix $A$, the eigenvalues of $A$ are denoted by $\lambda_1(A) \leq \cdots \leq \lambda_n(A)$, and the eigenvector associated with $\lambda_i(A)$ is denoted by $f_i(A)$ for $1 \leq i \leq n$.

**Hypergraphs.** Let $H = (V_H, E_H, w)$ be a hypergraph with $n = |V_H|$ vertices and weight function $w : E_H \mapsto \mathbb{R}^+$. For any vertex $v \in V_H$, the degree of $v$ is defined by $d_H(v) \triangleq \sum_{e \in E_H} w(e) \cdot \mathbb{I}[v \in e]$, where $\mathbb{I}[X] = 1$ if event $X$ holds and $\mathbb{I}[X] = 0$ otherwise. The rank of edge $e \in E_H$ is the total number of vertices in $e$. For any $A, B \subset V_H$, the cut value between $A$ and $B$ is defined by

$$w(A, B) \triangleq \sum_{e \in E_H} w(e) \cdot \mathbb{I}[e \cap A \neq \emptyset \wedge e \cap B \neq \emptyset].$$

Sometimes, we are required to analyse the weights of edges that intersect some vertex sets and not others. To this end, we define for any $A, B, C \subseteq V_H$ that

$$w(A, B \mid C) \triangleq \sum_{e \in E_H} w(e) \cdot \mathbb{I}[e \cap A \neq \emptyset \wedge e \cap B \neq \emptyset \wedge e \cap C = \emptyset],$$

and we sometimes write $w(A \mid C) \triangleq w(A, A \mid C)$ for simplicity. Generalising the notion of the bipartiteness ratio of a 2-graph, the *bipartiteness ratio* of sets $L, R$ in a hypergraph $H$ is defined by

$$\beta_H(L, R) \triangleq \frac{2w(L|\overline{L}) + 2w(R|\overline{R}) + w(L, \overline{L \cup R}|R) + w(R, \overline{L \cup R}|L)}{\mathrm{vol}(L \cup R)},$$

and we define $\beta_H \triangleq \min_{S \subset V} \beta_H(S, V \setminus S)$. For any hypergraph $H$ and $f \in \mathbb{R}^n$, we define the discrepancy of an edge $e \in E_H$ with respect to $f$ as

$$\Delta_f(e) \triangleq \max_{u \in e} f(u) + \min_{v \in e} f(v).$$

For any non-linear operator $J : \mathbb{R}^n \mapsto \mathbb{R}^n$, we say that $(\lambda, f)$ is an eigen-pair if and only if $Jf = \lambda f$ and note that in general, a non-linear operator can have any number of eigenvalues and eigenvectors.

It is important to remember that throughout the paper, we always use the letter $H$ to represent a hypergraph, and $G$ to represent a 2-graph.

**Clique reduction.** The *clique reduction* of a hypergraph $H$ is a 2-graph $G$ such that $V_G = V_H$ and for every edge $e \in E_H$, $G$ contains a clique on the vertices in $e$ with edge weights $1/(r_e - 1)$ where $r_e$ is the rank of the edge $e$. The clique reduction is a common tool for designing hypergraph algorithms [1, 7, 35], and for this reason we use it as a baseline algorithm in this paper. We note that hypergraph algorithms based on the clique reduction often perform less well when there are edges with large rank in the hypergraph. Specifically, in Appendix C we use two $r$-uniform hypergraphs as examples to show that no matter how we weight the edges in the clique reduction, some cuts cannot be approximated better than a factor of $O(r)$. This is one of the main reasons to develop spectral theory for hypergraphs through heat diffusion processes [5, 27, 33].

## 3   Diffusion process and the algorithm

In this section, we propose a new diffusion process in hypergraphs and use it to design a polynomial-time algorithm for finding bipartite components in hypergraphs. We first study 2-graphs to give some intuition, and then generalise to hypergraphs and describe our algorithm. Finally, we sketch some of the detailed analysis which proves that the diffusion process is well defined.

### 3.1   The diffusion process in $2$-graphs

To discuss the intuition behind our designed diffusion process, let us look at the case of 2-graphs. Let $G = (V, E, w)$ be a 2-graph, and we have for any $x \in \mathbb{R}^n$ that

$$\frac{x^{\intercal} \mathcal{J}_G x}{x^{\intercal} x} = \frac{x^{\intercal}(I + D_G^{-1/2} A_G D_G^{-1/2})x}{x^{\intercal} x}.$$

By setting $x = D_G^{1/2} y$, we have that

$$\frac{x^{\intercal} \mathcal{J}_G x}{x^{\intercal} x} = \frac{y^{\intercal} D_G^{1/2} \mathcal{J}_G D_G^{1/2} y}{y^{\intercal} D_G y} = \frac{y^{\intercal}(D_G + A_G)y}{y^{\intercal} D_G y} = \frac{\sum_{\{u,v\} \in E_G} w(u,v) \cdot (y(u) + y(v))^2}{\sum_{u \in V_G} d_G(u) \cdot y(u)^2}. \quad (1)$$

It is easy to see that $\lambda_1(\mathcal{J}_G) = 0$ if $G$ is bipartite, and it is known that $\lambda_1(\mathcal{J}_G)$ and its corresponding eigenvector $f_1(\mathcal{J}_G)$ are closely related to two densely connected components of $G$ [28]. Moreover, similar to the heat equation for graph Laplacians $L_G$, suppose $D_G f_t \in \mathbb{R}^n$ is some measure on the vertices of $G$, then a diffusion process defined by the differential equation

$$\frac{\mathrm{d}f_t}{\mathrm{d}t} = -D_G^{-1} J_G f_t \quad (2)$$

will converge to the minimum eigenvalue of $D_G^{-1} J_G$ and can be employed to find two densely connected components of the underlying 2-graph.[3]

### 3.2   The hypergraph diffusion and our algorithm

Now we study whether one can construct a new hypergraph operator $J_H$ which generalises the diffusion in 2-graphs to hypergraphs. First of all, we focus on a fixed time $t$ with measure vector $D_H f_t \in \mathbb{R}^n$ and ask whether we can follow (2) and define the rate of change

$$\frac{\mathrm{d}f_t}{\mathrm{d}t} = -D_H^{-1} J_H f_t$$

so that the diffusion can proceed for an infinitesimal time step. Our intuition is that the rate of change due to some edge $e \in E_H$ should involve only the vertices in $e$ with the maximum or minimum value in the normalised measure $f_t$. To formalise this, for any edge $e \in E_H$, we define

$$S_f(e) \triangleq \{v \in e : f_t(v) = \max_{u \in e} f_t(u)\} \quad \text{and} \quad I_f(e) \triangleq \{v \in e : f_t(v) = \min_{u \in e} f_t(u)\}.$$

---

[3]For the reader familiar with the heat diffusion process of 2-graphs (e.g., [9, 17]), we remark that the above-defined process essentially employs the operation $J_G$ to replace the Laplacian $L_G$ when defining the heat diffusion: through $J_G$, the heat diffusion can be used to find two densely connected components of $G$.

That is, for any edge $e$ and normalised measure $f_t$, $S_f(e) \subseteq e$ consists of the vertices $v$ adjacent to $e$ whose $f_t(v)$ values are maximum and $I_f(e) \subseteq e$ consists of the vertices $v$ adjacent to $e$ whose $f_t(v)$ values are minimum. See Figure 2 for an example. Then, applying the $J_H$ operator to a vector $f_t$ should be equivalent to applying the operator $J_G$ for some 2-graph $G$ which we construct by splitting the weight of each hyperedge $e \in E_H$ between the edges in $S_f(e) \times I_f(e)$. Similar to the case for 2-graphs and (1), for any $x = D_H^{1/2} f_t$ this will give us the quadratic form

$$\frac{x^\mathsf{T} D_H^{-1/2} J_H D_H^{-1/2} x}{x^\mathsf{T} x} = \frac{f_t^\mathsf{T} J_G f_t}{f_t^\mathsf{T} D_H f_t} = \frac{\sum_{\{u,v\} \in E_G} w_G(u,v) \cdot (f_t(u) + f_t(v))^2}{\sum_{u \in V_G} d_H(u) \cdot f_t(u)^2}$$
$$= \frac{\sum_{e \in E_H} w_H(e)(\max_{u \in e} f_t(u) + \min_{v \in e} f_t(v))^2}{\sum_{u \in V_H} d_H(u) \cdot f_t(u)^2},$$

where $w_G(u,v)$ is the weight of the edge $\{u,v\}$ in $G$, and $w_H(e)$ is the weight of the edge $e$ in $H$. We will show in the proof of Theorem 1 that $J_H$ has an eigenvalue of $0$ if the hypergraph is 2-colourable[4], and that the spectrum of $J_H$ is closely related to the hypergraph bipartiteness.

For this reason, we would expect that the diffusion process based on the operator $J_H$ can be used to find sets with small hypergraph bipartiteness. However, one needs to be very cautious here as, by the nature of the diffusion process, the values $f_t(v)$ of all the vertices $v$ change over time and, as a result, the sets $S_f(e)$ and $I_f(e)$ that consist of the vertices with the maximum and minimum $f_t$-value might change after an *infinitesimal* time step; this will prevent the process from continuing. We will discuss this issue in detail through the so-called *Diffusion Continuity Condition* in Section 3.3. In essence, the diffusion continuity condition ensures that one can always construct a 2-graph $G$ by allocating the weight of each hyperedge $e$ to the edges in $S_f(e) \times I_f(e)$ such that the sets $S_f(e)$ and $I_f(e)$ will not change in infinitesimal time although $f_t$ changes according to $(\mathrm{d}f_t)/(\mathrm{d}t) = -D_H^{-1} J_G f_t$. We will also present an efficient

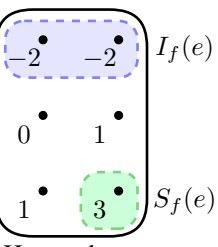

Figure 2: Illustration of $S_f(e)$ and $I_f(e)$. Vertices are labelled with their value in $f_t$.

procedure in Section 3.3 to compute the weights of edges in $S_f(e) \times I_f(e)$. All of these guarantee that (i) every 2-graph that corresponds to the hypergraph diffusion process at any time step can be efficiently constructed; (ii) with this sequence of constructed 2-graphs, the diffusion process defined by $J_H$ is able to continue until the heat distribution converges. With this, we summarise the main idea of our presented algorithm as follows:

- First of all, we introduce some arbitrary $f_0 \in \mathbb{R}^n$ as the initial diffusion vector, and a step size parameter $\epsilon > 0$ to discretise the diffusion process. At each step, the algorithm constructs the 2-graph $G$ guaranteed by the diffusion continuity condition, and updates $f_t \in \mathbb{R}^n$ according to the rate of change $(\mathrm{d}f_t)/(\mathrm{d}t) = -D_H^{-1} J_G f_t$. The algorithm terminates when $f_t$ has converged, i.e., the ratio between the current Rayleigh quotient $(f_t^\mathsf{T} J_G f_t)/(f_t^\mathsf{T} D_H f_t)$ and the one in the previous time step is bounded by some predefined constant.
- Secondly, similar to many previous spectral graph clustering algorithms (e.g. [3, 27, 28]), the algorithm constructs the sweep sets defined by $f_t$ and returns the two sets with minimum $\beta_H$-value among all the constructed sweep sets. Specifically, for every $1 \leq i \leq n$, the algorithm constructs $L_j = \{v_i : |f_t(v_i)| \geq |f_t(v_j)| \wedge f_t(v_i) < 0\}$ and $R_j = \{v_i : |f_t(v_i)| \geq |f_t(v_j)| \wedge f_t(v_i) \geq 0\}$. Then, between the $n$ pairs $(L_j, R_j)$, the algorithm returns the one with the minimum $\beta_H$-value.

See Algorithm 1 for the formal description, and its performance is summarised in Theorem 1.

**Theorem 1** (Main Result). *Given a hypergraph $H = (V_H, E_H, w)$ and parameter $\epsilon > 0$, the following holds:*

1. *There is an algorithm that finds an eigen-pair $(\lambda, f)$ of the operator $J_H$ such that $\lambda \leq \lambda_1(J_G)$, where $G$ is the clique reduction of $H$ and the inequality is strict if $\min_{e \in E_H} r_e > 2$ where $r_e$ is the rank of $e$. The algorithm runs in $\mathrm{poly}(|V_H|, |E_H|, 1/\epsilon)$ time.*

---

[4]Hypergraph $H$ is 2-colourable if there are disjoint sets $L, R \subset V_H$ such that every edge intersects $L$ and $R$.

2. *Given an eigen-pair $(\lambda, f)$ of the operator $J_H$, there is an algorithm that constructs the two-sided sweep sets defined on $f$, and finds sets $L$ and $R$ such that $\beta_H(L, R) \leq \sqrt{2\lambda}$. The algorithm runs in $\mathrm{poly}(|V_H|, |E_H|)$ time.*

---

**Algorithm 1:** FINDBIPARTITECOMPONENTS

**Input** : Hypergraph $H$, starting vector $f_0 \in \mathbb{R}^n$, step size $\epsilon > 0$
**Output** : Sets $L$ and $R$
$t := 0$
**while** $f_t$ *has not converged* **do**
$\quad$ Use $f_t$ to construct 2-graph $G$ satisfying the diffusion continuity condition
$\quad f_{t+\epsilon} := f_t - \epsilon D_H^{-1} J_G f_t$
$\quad t := t + \epsilon$
**end**
Set $j := \arg\min_{1 \leq i \leq n} \beta_H(L_i, R_i)$
**return** $(L_j, R_j)$

---

**Remark 1.** *We make the important remark that there is no polynomial-time algorithm which guarantees any multiplicative approximation of the minimum hypergraph bipartiteness value $\beta_H$, unless $\mathrm{P} = \mathrm{NP}$. We prove this in Appendix C by a reduction from the NP-complete HYPERGRAPH 2-COLOURABILITY problem. This means that the problem we consider in this work is fundamentally more difficult than the equivalent problem for 2-graphs, as well as the problem of finding a sparse cut in a hypergraph. For this reason, the analysis of the non-linear hypergraph Laplacian operator [5, 27] cannot be applied in our case.*

### 3.3 Dealing with the diffusion continuity condition

It remains for us to discuss the diffusion continuity condition, which guarantees that $S_f(e)$ and $I_f(e)$ will not change in infinitesimal time and the diffusion process will eventually converge to some stable distribution. Formally, let $f_t$ be the normalised measure on the vertices of $H$, and let

$$r \triangleq \frac{\mathrm{d}f_t}{\mathrm{d}t} = -D_H^{-1} J_H f_t$$

be the derivative of $f_t$, which describes the rate of change for every vertex at the current time $t$. We write $r(v)$ for any $v \in V_H$ as $r(v) = \sum_{e \in E_H} r_e(v)$, where $r_e(v)$ is the contribution of edge $e$ towards the rate of change of $v$. Now we discuss three rules that we expect the diffusion process to satisfy, and later prove that these three rules uniquely define the rate of change $r$.

First of all, as we mentioned in Section 3.2, we expect that only the vertices in $S_f(e) \cup I_f(e)$ will participate in the diffusion process, i.e., $r_e(u) = 0$ unless $u \in S_f(e) \cup I_f(e)$. Moreover, any vertex $u$ participating in the diffusion process must satisfy the following:

- Rule (0a): if $|r_e(u)| > 0$ and $u \in S_f(e)$, then $r(u) = \max_{v \in S_f(e)}\{r(v)\}$.
- Rule (0b): if $|r_e(u)| > 0$ and $u \in I_f(e)$, then $r(u) = \min_{v \in I_f(e)}\{r(v)\}$.

To explain Rule (0), notice that for an infinitesimal time, $f_t(u)$ will be increased according to $(\mathrm{d}f_t/\mathrm{d}t)(u) = r(u)$. Hence, by Rule (0) we know that, if $u \in S_f(e)$ (resp. $u \in I_f(e)$) participates in the diffusion process in edge $e$, then in an infinitesimal time $f(u)$ will remain the maximum (resp. minimum) among the vertices in $e$. Such a rule is necessary to ensure that the vertices involved in the diffusion in edge $e$ do not change in infinitesimal time, and the diffusion process is able to continue.

Our next rule states that the total rate of change of the measure due to edge $e$ is equal to $-w(e) \cdot \Delta_f(e)$:

- Rule (1): $\sum_{v \in S_f(e)} d(v) r_e(v) = \sum_{v \in I_f(e)} d(v) r_e(v) = -w(e) \cdot \Delta_f(e)$ for all $e \in E_H$.

This rule is a generalisation from the operator $J_G$ in 2-graphs. In particular, since $D_G^{-1} J_G f_t(u) = \sum_{\{u,v\} \in E_G} w_G(u, v)(f_t(u) + f_t(v))/d_G(u)$, the rate of change of $f_t(u)$ due to the edge $\{u, v\} \in E_G$ is $-w_G(u, v)(f_t(u) + f_t(v))/d_G(u)$. Rule (1) states that in the hypergraph case the rate of change of the vertices in $S_f(e)$ and $I_f(e)$ together behave like the rate of change of $u$ and $v$ in the 2-graph case.

One might have expected that these two rules together will define a unique process. Unfortunately, this isn't the case and we present a counterexample in Appendix A. To overcome this, we introduce the following stronger rule to replace Rule (0):

- Rule (2a): Assume that $|r_e(u)| > 0$ and $u \in S_f(e)$.

    - If $\Delta_f(e) > 0$, then $r(u) = \max_{v \in S_f(e)}\{r(v)\}$;
    - If $\Delta_f(e) < 0$, then $r(u) = r(v)$ for all $v \in S_f(e)$.

- Rule (2b): Assume that $|r_e(u)| > 0$ and $u \in I_f(e)$:

    - If $\Delta_f(e) < 0$, then $r(u) = \min_{v \in I_f(e)}\{r(v)\}$;
    - If $\Delta_f(e) > 0$, then $r(u) = r(v)$ for all $v \in I_f(e)$.

Notice that the first conditions of Rules (2a) and (2b) correspond to Rules (0a) and (0b) respectively; the second conditions are introduced for purely technical reasons: they state that, if the discrepancy of $e$ is negative (resp. positive), then all the vertices $u \in S_f(e)$ (resp. $u \in I_f(e)$) will have the same value of $r(u)$. Theorem 2 shows that there is a unique $r \in \mathbb{R}^n$ that satisfies Rules (1) and (2), and $r$ can be computed in polynomial time. Therefore, our two rules uniquely define a diffusion process, and we can use the computed $r$ to simulate the continuous diffusion process with a discretised version.[5]

**Theorem 2.** *For any given $f_t \in \mathbb{R}^n$, there is a unique $r = \mathrm{d}f_t/\mathrm{d}t$ and associated $\{r_e(v)\}_{e \in E, v \in V}$ that satisfy Rule (1) and (2), and $r$ can be computed in polynomial time by linear programming.*

**Remark 2.** *The rules we define and the proof of Theorem 2 are more involved than those used in [5] to define the hypergraph Laplacian operator. In particular, in contrast to [5], in our case the discrepancy $\Delta_f(e)$ within a hyperedge $e$ can be either positive or negative. This results in the four different cases in Rule (2) which must be carefully considered throughout the proof of Theorem 2.*

## 4 Experiments

In this section, we evaluate the performance of our new algorithm on synthetic and real-world datasets. All algorithms are implemented in Python 3.6, using the `scipy` library for sparse matrix representations and linear programs. The experiments are performed using an Intel(R) Core(TM) i5-8500 CPU @ 3.00GHz processor, with 16 GB RAM. Our code can be downloaded from https://github.com/pmacg/hypergraph-bipartite-components.

Since ours is the first proposed algorithm for approximating hypergraph bipartiteness, we will compare it to a simple and natural baseline algorithm, which we call CLIQUECUT (CC). In this algorithm, we construct the clique reduction of the hypergraph and use the two-sided sweep-set algorithm described in [28] to find a set with low bipartiteness in the clique reduction.[6]

Additionally, we will compare two versions of our proposed algorithm. FINDBIPARTITECOMPONENTS (FBC) is our new algorithm described in Algorithm 1 and FBCAPPROX (FBCA) is an approximate version in which we do not solve the linear programs in Theorem 2 to compute the graph $G$. Instead, at each step of the algorithm, we construct $G$ by splitting the weight of each hyperedge $e$ evenly between the edges in $S(e) \times I(e)$.

We always set the parameter $\epsilon = 1$ for FBC and FBCA, and we set the starting vector $f_0 \in \mathbb{R}^n$ for the diffusion to be the eigenvector corresponding to the minimum eigenvalue of $J_G$, where $G$ is the clique reduction of the hypergraph $H$.

### 4.1 Synthetic datasets

We first evaluate the algorithms using a random hypergraph model. Given the parameters $n$, $r$, $p$, and $q$, we generate an $n$-vertex $r$-uniform hypergraph in the following way: the vertex set $V$ is divided

---

[5]Note that the graph $G$ used for the diffusion at time $t$ can be easily computed from the $\{r_e(v)\}$ values, although in practice this is not actually needed since the $r(u)$ values can be used to update the diffusion directly.

[6]We choose to use the algorithm in [28] here since, as far we know, this is the only non-SDP based algorithm for solving the MAX-CUT problem for 2-graphs. Notice that, although SDP-based algorithms achieve a better approximation ratio for the MAX-CUT problem, they are not practical even for hypergraphs of medium sizes.

into two clusters $L$ and $R$ of size $n/2$. For every set $S \subset V$ with $|S| = r$, if $S \subset L$ or $S \subset R$ we add the hyperedge $S$ with probability $p$ and otherwise we add the hyperedge with probability $q$. We remark that this is a special case of the hypergraph stochastic block model (e.g., [6]). We limit the number of free parameters for simplicity while maintaining enough flexibility to generate random hypergraphs with a wide range of optimal $\beta_H$-values.

We will compare the algorithms' performance using four metrics: the hypergraph bipartiteness ratio $\beta_H(L, R)$, the clique graph bipartiteness ratio $\beta_G(L, R)$, the $F_1$-score [30] of the returned clustering, and the runtime of the algorithm. Throughout this subsection, we always report the average result on 10 hypergraphs randomly generated with each parameter configuration.

**Comparison of FBC and FBCA.** We first fix the values $n = 200$, $r = 3$, and $p = 10^{-4}$ and vary the ratio of $q/p$ from 2 to 6 which produces hypergraphs with 250 to 650 edges. The performance of each algorithm on these hypergraphs is shown in Figure 3 from which we can make the following observations:

- From Figure 3 (a) we observe that FBC and FBCA find sets with very similar bipartiteness and they perform better than the CLIQUECUT baseline.
- From Figure 3 (b) we can see that our proposed algorithms produce output with a lower $\beta_G$-value than the output of the CLIQUECUT algorithm. This is a surprising result given that CLIQUECUT operates directly on the clique graph.
- Figure 3 (c) shows that the FBCA algorithm is much faster than FBC.

From these observations, we conclude that in practice it is sufficient to use the much faster FBCA algorithm in place of the FBC algorithm.

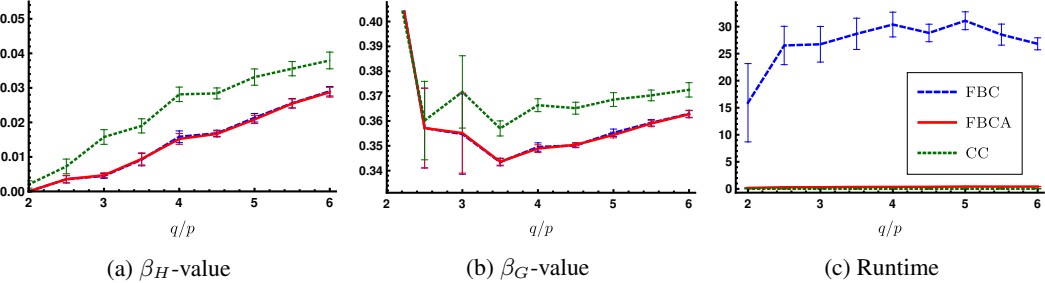

(a) $\beta_H$-value      (b) $\beta_G$-value      (c) Runtime

Figure 3: The average performance and standard error of each algorithm when $n = 200$, $r = 3$ and $p = 10^{-4}$.

**Experiments on larger graphs.** We now compare only the FBCA and CLIQUECUT algorithms, which allows us to run on hypergraphs with higher rank and number of vertices. We fix the parameters $n = 2000$, $r = 5$, and $p = 10^{-11}$, producing hypergraphs with between 5000 and 75000 edges[7] and show the results in Figure 4. Our algorithm consistently and significantly outperforms the baseline on every metric and across a wide variety of input hypergraphs.

To compare the algorithms' runtime, we fix the parameter $n = 2000$ and the ratio $q = 2p$ and report the runtime of the FBCA and CC algorithms on a variety of hypergraphs in Table 1. Our proposed algorithm takes more time than

Table 1: The runtime in seconds of the FBCA and CC algorithms.

| | | | Avg. Runtime | |
|---|---|---|---|---|
| $r$ | $p$ | Avg. $|E_H|$ | FBCA | CC |
| | $10^{-9}$ | 1239 | 1.15 | 0.12 |
| 4 | $10^{-8}$ | 12479 | 10.14 | 0.86 |
| | $10^{-7}$ | 124717 | 89.92 | 9.08 |
| | $10^{-11}$ | 5177 | 3.99 | 0.62 |
| 5 | $10^{-10}$ | 51385 | 44.10 | 6.50 |
| | $10^{-9}$ | 514375 | 368.48 | 69.25 |

[7]In our model, a very small value of $p$ and $q$ is needed since in an $n$-vertex, $r$-uniform hypergraph there are $\binom{n}{r}$ possible edges which can be a very large number. In this case, $\binom{2000}{5} \approx 2.6 \times 10^{14}$.

the baseline CC algorithm but both appear to scale linearly in the size of the input hypergraph[8] which suggests that our algorithm's runtime is roughly a constant factor multiple of the baseline.

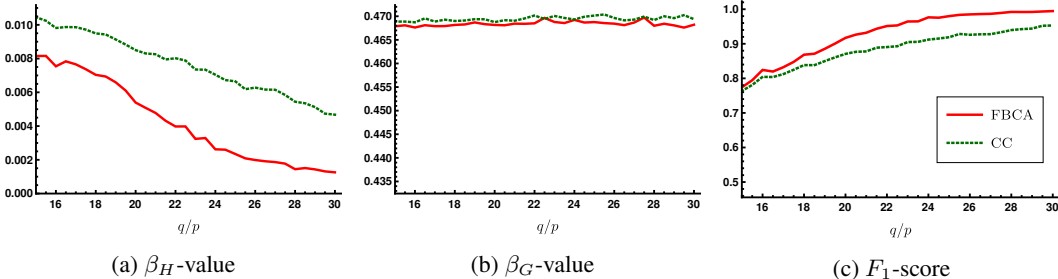

(a) $\beta_H$-value    (b) $\beta_G$-value    (c) $F_1$-score

Figure 4: The average performance of each algorithm when $n = 2000$, $r = 5$, and $p = 10^{-11}$. We omit the error bars because they are too small to read.

## 4.2 Real-world datasets

Next, we demonstrate the broad utility of our algorithm on complex real-world datasets with higher-order relationships which are most naturally represented by hypergraphs. Moreover, the hypergraphs are *inhomogeneous*, meaning that they contain vertices of different types, although this information is not available to the algorithm and so an algorithm has to treat every vertex identically. We demonstrate that our algorithm is able to find clusters which correspond to the vertices of different types. Table 2 shows the $F_1$-score of the clustering produced by our algorithm on each dataset and demonstrates that it consistently outperforms the CLIQUECUT algorithm.

**Penn Treebank.** The Penn Treebank dataset is an English-language corpus with examples of written American English from several sources, including fiction and journalism [22]. The dataset contains $49,208$ sentences and over $1$ million words, which are labelled with their part of speech. We construct a hypergraph in the following way: the vertex set consists of all the verbs, adverbs, and adjectives which occur at least $10$ times in the corpus, and for every $4$-gram (a sequence of $4$ words) we add a hyperedge containing the co-occurring words. This results in a hypergraph with $4,686$ vertices and $176,286$ edges. The clustering returned by our algorithm correctly distinguishes between verbs and non-verbs with an accuracy of 67%. This experiment demonstrates that our *unsupervised* general purpose algorithm is capable of recovering non-trivial structure in a dataset which would ordinarily be clustered using significant domain knowledge, or a complex pre-trained model [2, 16].

Table 2: The performance of the FBCA and CC algorithms on real world datasets.

|  |  | $F_1$-Score | |
| --- | --- | --- | --- |
| Dataset | Cluster | FBCA | CC |
| Penn | Verbs | **0.73** | 0.69 |
| Treebank | Non-Verbs | **0.59** | 0.56 |
| DBLP | Conferences | **1.00** | 0.25 |
|  | Authors | **1.00** | 0.98 |

**DBLP.** We construct a hypergraph from a subset of the DBLP network consisting of $14,376$ papers published in artificial intelligence and machine learning conferences [12, 32]. For each paper, we include a hyperedge linking the authors of the paper with the conference in which it was published, giving a hypergraph with $14,495$ vertices and $14,376$ edges. The clusters returned by our algorithm successfully separate the authors from the conferences with an accuracy of 100%.

## 5 Concluding remarks

In this paper, we introduce a new hypergraph Laplacian-type operator and apply this operator to design an algorithm that finds almost bipartite components in hypergraphs. Our experimental results

---

[8]Although $n$ is fixed, the CLIQUECUT algorithm's runtime is not constant since the time to compute an eigenvalue of the sparse adjacency matrix scales with the number and rank of the hyperedges.

demonstrate the potentially wide applications of spectral hypergraph theory, and so we believe that designing faster spectral hypergraph algorithms is an important future research direction in algorithms and machine learning. This will allow spectral hypergraph techniques to be applied more effectively to analyse the complex datasets which occur with increasing frequency in the real world.

## Acknowledgements

Peter Macgregor is supported by the Langmuir PhD Scholarship, and He Sun is supported by an EPSRC Early Career Fellowship (EP/T00729X/1).

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
