# Finding Bipartite Components in Hypergraphs
## Supplementary Material

## Contents

## A  Proof of Theorem 2

In this section we prove Theorem 2. After giving some additional preliminaries, and discussing the rules of the diffusion process, we will construct a linear program which can compute the rate of change $r$ satisfying the rules of the diffusion process. We then give a complete analysis of the new linear program which establishes Theorem 2.

### A.1  Additional preliminaries

Given a hypergraph $H = (V_H, E_H, w)$, and a weighted measure vector $f_t$, the discrepancy of $H$ with respect to $f_t$ is defined by

$$D(f_t) \triangleq \frac{\sum_{e \in E_H} w(e) \cdot \Delta_{f_t}(e)^2}{\sum_{v \in V_H} d_H(v) \cdot f_t(v)^2}.$$

Additionally, we define the Rayleigh quotient of the operator $J_H$ as

$$R_H(f_t) \triangleq \frac{f_t^\mathsf{T} J_H f_t}{f_t^\mathsf{T} D_H f_t}.$$

For any edge $e \in E_H$ and vector $f_t$, let

$$c_{f_t}(e) \triangleq w(e) \left| \Delta_{f_t}(e) \right|$$

35th Conference on Neural Information Processing Systems (NeurIPS 2021)

and for any set $S \subseteq E_H$, let

$$c_{f_t}(S) \triangleq \sum_{e \in S} c_{f_t}(e).$$

For any $f, g \in \mathbb{R}^n$, the weighted inner product between $f$ and $g$ is defined by $\langle f, g \rangle_w \triangleq f^\intercal D_H g$, where $D_H \in \mathbb{R}^{n \times n}$ is the diagonal matrix consisting of the degrees of all the vertices of H. The weighted norm of $f$ is defined by $\|f\|_w^2 \triangleq \langle f, f \rangle_w$.

## A.2   Counter-example showing that rule (2) is needed

First, we recall the rules which the rate of change of the diffusion process must satisfy. Given a hypergraph $H = (V_H, E_H, w)$ and a weighted diffusion vector $f_t$, let

$$r = \frac{\mathrm{d}f_t}{\mathrm{d}t}$$

be the rate of change of the vector $f_t$. Additionally, recall that for any vertex $v$, we write

$$r(v) = \sum_{e \in E_H} r_e(v),$$

where $r_e(v)$ is the contribution of the edge $e$ to the rate of change of $f_t(v)$. Then, we define the rules of the diffusion as follows:

- Rule (0a): if $|r_e(u)| > 0$ and $u \in S_{f_t}(e)$, then $r(u) = \max_{v \in S_{f_t}(e)}\{r(v)\}$.

- Rule (0b): if $|r_e(u)| > 0$ and $u \in I_{f_t}(e)$, then $r(u) = \min_{v \in I_{f_t}(e)}\{r(v)\}$.

- Rule (1): $\sum_{v \in S_{f_t}(e)} d(v)r_e(v) = \sum_{v \in I_{f_t}(e)} d(v)r_e(v) = -w(e) \cdot \Delta_{f_t}(e)$ for all $e \in E_H$.

- Rule (2a): Assume that $|r_e(u)| > 0$ and $u \in S_{f_t}(e)$.

  - If $\Delta_{f_t}(e) > 0$, then $r(u) = \max_{v \in S_{f_t}(e)}\{r(v)\}$;
  - If $\Delta_{f_t}(e) < 0$, then $r(u) = r(v)$ for all $v \in S_{f_t}(e)$.

- Rule (2b): Assume that $|r_e(u)| > 0$ and $u \in I_{f_t}(e)$:

  - If $\Delta_{f_t}(e) < 0$, then $r(u) = \min_{v \in I_{f_t}(e)}\{r(v)\}$;
  - If $\Delta_{f_t}(e) > 0$, then $r(u) = r(v)$ for all $v \in I_{f_t}(e)$.

One might have expected that Rules (0) and (1) together would define a unique process. Unfortunately, this isn't the case. For example, let us define an unweighted hypergraph $H = (V_H, E_H)$, where $V_H = \{u, v, w\}$ and $E_H = \{\{u, v, w\}\}$. By setting the measure to be $f_t = (1, 1, -2)^\intercal$ and $e = \{u, v, w\}$, we have $\Delta_{f_t}(e) = -1$, and $r(u) + r(v) = r(w) = 1$ by Rule (1). In such a scenario, either $\{u, w\}$ or $\{v, w\}$ can participate in the diffusion and satisfy Rule (0), which makes the process not uniquely defined and so we introduce Rule (2) to ensure that there will be a unique vector $r$ which satisfies the rules.

## A.3   Computing $r$ by a linear program

Now we present an algorithm that computes the vector $r = \mathrm{d}f_t/\mathrm{d}t$ for any $f_t \in \mathbb{R}^n$. Without loss of generality, let us fix $f_t \in \mathbb{R}^n$, and define a set of equivalence classes $\mathcal{U}$ on V such that vertices $u, v \in V_H$ are in the same equivalence class if $f_t(u) = f_t(v)$. Next we study every equivalence class $U \in \mathcal{U}$ in turn, and will set the $r$-value of the vertices in $U$ recursively. In each iteration, we fix the $r$-value of some subset $P \subseteq U$ and recurse on $U \setminus P$. As we'll prove later, it's important to highlight that the recursive procedure ensures that the $r$-values assigned to the vertices always *decrease* after each recursion. Notice that it suffices to consider the edges $e$ in which $(S_{f_t}(e) \cup I_{f_t}(e)) \cap U \neq \emptyset$, since the diffusion process induced by other edges $e$ will have no impact on $r(u)$ for any $u \in U$. Hence, we introduce the sets

$$\mathcal{S}_U \triangleq \{e \in E_U : S_{f_t}(e) \cap U \neq \emptyset\}, \qquad \mathcal{I}_U \triangleq \{e \in E_U : I_{f_t}(e) \cap U \neq \emptyset\},$$

where $E_U$ consists of the edges adjacent to some vertex in $U$. To work with the four cases listed in Rule (2a) and (2b), we define

$$
\begin{aligned}
\mathcal{S}_U^+ &\triangleq \{e \in \mathcal{S}_U : \Delta_{f_t}(e) < 0\} \\
\mathcal{S}_U^- &\triangleq \{e \in \mathcal{S}_U : \Delta_{f_t}(e) > 0\} \\
\mathcal{I}_U^+ &\triangleq \{e \in \mathcal{I}_U : \Delta_{f_t}(e) < 0\} \\
\mathcal{I}_U^- &\triangleq \{e \in \mathcal{I}_U : \Delta_{f_t}(e) > 0\}
\end{aligned}
$$

Our objective is to find some $P \subseteq U$ and assign the same $r$-value to every vertex in $P$. To this end, for any $P \subseteq U$ we define

$$
\begin{aligned}
\mathcal{S}_{U,P}^+ &\triangleq \left\{e \in \mathcal{S}_U^+ : S_{f_t}(e) \subseteq P\right\}, \\
\mathcal{I}_{U,P}^+ &\triangleq \left\{e \in \mathcal{I}_U^+ : I_{f_t}(e) \subseteq P\right\}, \\
\mathcal{S}_{U,P}^- &\triangleq \left\{e \in \mathcal{S}_U^- : S_{f_t}(e) \cap P \neq \emptyset\right\}, \\
\mathcal{I}_{U,P}^- &\triangleq \{e \in \mathcal{I}_U^- : I_{f_t}(e) \subseteq P\}.
\end{aligned}
$$

These are the edges which will contribute to the rate of change of the vertices in $P$. Before continuing the analysis, we briefly explain the intuition behind these four definitions: (i) For $\mathcal{S}_{U,P}^+$, since every $e \in \mathcal{S}_U^+$ satisfies $\Delta_{f_t}(e) < 0$ and all the vertices in $S_{f_t}(e)$ must have the same value by Rule (2a), all such $e \in \mathcal{S}_{U,P}^+$ must satisfy that $S_{f_t}(e) \subseteq P$, since the unassigned vertices will receive lower values of $r$ in the remaining part of the recursion process. (ii) For $\mathcal{I}_{U,P}^+$, since every $e \in \mathcal{I}_U^+$ satisfies $\Delta_{f_t}(e) < 0$, Rule (2b) implies that if $r_e(u) \neq 0$ then $r(u) \leq r(v)$ for all $v \in I_{f_t}(e)$. Since unassigned vertices will receive lower values of $r$ later, such $e \in \mathcal{I}_{U,P}^+$ must satisfy $I_{f_t}(e) \subseteq P$. (iii) For $\mathcal{S}_{U,P}^-$, since every $e \in \mathcal{S}_U^-$ satisfies $\Delta_{f_t}(e) > 0$, by Rule (2a) it suffices that some vertex in $S_{f_t}(e)$ receives the assignment in the current iteration, i.e., every such $e$ must satisfy $S_{f_t}(e) \cap P \neq \emptyset$. (iv) The case for $\mathcal{I}_P^-$ is the same as $\mathcal{S}_P^+$.

As we expect all the vertices $u \in S_{f_t}(e)$ to have the same $r$-value for every $e$ as long as $\Delta_{f_t}(e) < 0$ by Rule (2a) and at the moment we are only considering the assignment of the vertices in $P$, we expect that

$$
\left\{e \in \mathcal{S}_U^+ \setminus \mathcal{S}_{U,P}^+ : S_{f_t}(e) \cap P \neq \emptyset\right\} = \emptyset, \tag{3}
$$

and this will ensure that, as long as $\Delta_{f_t}(e) < 0$ and some $u \in S_{f_t}(e)$ gets its $r$-value, then all the other vertices in $S_{f_t}(e)$ would be assigned the same value as $u$. Similarly, by Rule (2b), we expect all the vertices $u \in I_{f_t}(e)$ to have the same $r$-value for every $e$ as long as $\Delta_{f_t}(e) > 0$, and so we expect that

$$
\left\{e \in \mathcal{I}_U^- \setminus \mathcal{I}_{U,P}^- : I_{f_t}(e) \cap P \neq \emptyset\right\} = \emptyset. \tag{4}
$$

We will set the $r$-value by dividing the total discrepancy of the edges in $\mathcal{I}_{U,P}^+ \cup \mathcal{S}_{U,P}^+ \cup \mathcal{I}_{U,P}^- \cup \mathcal{S}_{U,P}^-$ between the vertices in $P$. As such, we would like to find some $P \subseteq U$ that maximises the value of

$$
\frac{1}{\text{vol}(P)} \cdot \left( \sum_{e \in \mathcal{S}_{U,P}^+ \cup \mathcal{I}_{U,P}^+} c_{f_t}(e) - \sum_{e \in \mathcal{S}_{U,P}^- \cup \mathcal{I}_{U,P}^-} c_{f_t}(e) \right).
$$

Taking all of these requirements into account, we will show that, for any equivalence set $U$, we can find the desired set $P$ by solving the following linear program:

$$\text{maximise} \quad c(x) = \sum_{e \in \mathcal{S}_U^+ \cup \mathcal{I}_U^+} c_{f_t}(e) \cdot x_e - \sum_{e \in \mathcal{S}_U^- \cup \mathcal{I}_U^-} c_{f_t}(e) \cdot x_e \tag{5}$$

$$\text{subject to} \quad \sum_{v \in U} d(v) y_v = 1$$

$$
\begin{aligned}
x_e = y_u \qquad & e \in \mathcal{S}_U^+, u \in S_{f_t}(e), \\
x_e \leq y_u \qquad & e \in \mathcal{I}_U^+, u \in I_{f_t}(e), \\
x_e \geq y_u \qquad & e \in \mathcal{S}_U^-, u \in S_{f_t}(e), \\
x_e = y_u \qquad & e \in \mathcal{I}_U^-, u \in I_{f_t}(e), \\
x_e, y_v \geq 0 \qquad & \forall v \in U, e \in E_U.
\end{aligned}
$$

Since the LP only gives partial assignment to the vertices' $r$-values, we solve the same LP on the reduced instance given by the set $U \setminus P$. The formal description of our algorithm is given in Algorithm 2.

---

**Algorithm 2:** ComputeChangeRate$(U, E_U)$

---

**Input** : vertex set $U \subseteq V$, and edge set $E_U$
**Output** : Values of $\{r(v)\}_{v \in U}$
Construct sets $\mathcal{S}_U^+, \mathcal{S}_U^-, \mathcal{I}_U^+$, and $\mathcal{I}_U^-$
Solve the linear program defined by (5), and define $P := \{v \in U : y(v) > 0\}$
Construct sets $\mathcal{S}_{U,P}^+, \mathcal{S}_{U,P}^-, \mathcal{I}_{U,P}^+$, and $\mathcal{I}_{U,P}^-$
Set $C(P) := c_{f_t}\left(\mathcal{S}_{U,P}^+\right) + c_{f_t}\left(\mathcal{I}_{U,P}^+\right) - c_{f_t}\left(\mathcal{S}_{U,P}^-\right) - c_{f_t}\left(\mathcal{I}_{U,P}^-\right)$
Set $\delta(P) := C(P)/\text{vol}(P)$
Set $r(u) := \delta(P)$ for every $u \in P$
ComputeChangeRate$\left(U \setminus P, E_U \setminus \left(\mathcal{S}_{U,P}^+ \cup \mathcal{I}_{U,P}^+ \cup \mathcal{S}_{U,P}^- \cup \mathcal{I}_{U,P}^-\right)\right)$

---

### A.4 Analysis of the linear program

Now we analyse Algorithm 2, and the properties of the $r$-values it computes. Specifically, we will show the following facts which will together allow us to establish Theorem 2.

1. Algorithm 2 always produces a unique vector $r$, no matter which optimal result is returned when computing the linear program (5).
2. If there is any vector $r$ which is consistent with Rules (1) and (2), then it must be equal to the output of Algorithm 2.
3. The vector $r$ produced by Algorithm 2 is consistent with Rules (1) and (2).

**The output of Algorithm 2 is unique.** First of all, for any $P \subseteq U$ that satisfies (3) and (4), we define vectors $x_P$ and $y_P$ by

$$x_P(e) = \begin{cases} \frac{1}{\text{vol}(P)} & \text{if } e \in \mathcal{S}_{U,P}^+ \cup \mathcal{I}_{U,P}^+ \cup \mathcal{S}_{U,P}^- \cup \mathcal{I}_{U,P}^- \\ 0 & \text{otherwise} \end{cases},$$

$$y_P(v) = \begin{cases} \frac{1}{\text{vol}(P)} & \text{if } v \in P \\ 0 & \text{otherwise} \end{cases},$$

and $z_P = (x_P, y_P)$. It is easy to verify that $(x_P, y_P)$ is a feasible solution to (5) with the objective value $c(x_P) = \delta(P)$. We will prove that ComputeChangeRate (Algorithm 2) computes a unique vector $r$ regardless of how ties are broken when computing the subsets $P$.

For any feasible solution $z = (x, y)$, we say that a non-empty set $Q$ is a *level set* of $z$ if there is some $t > 0$ such that $Q = \{u \in U : y_u \geq t\}$. We'll first show that any non-empty level set of an optimal solution $z$ corresponds to an optimal solution.

**Lemma 1.** *Suppose that $z^\star = (x^\star, y^\star)$ is an optimal solution of the linear program (5). Then, any non-empty level set $Q$ of $z^\star$ corresponds to an optimal solution of (5) as well.*

*Proof.* Let $P = \{v \in V_H : y^\star(v) > 0\}$. The proof is by case distinction. We first look at the case in which all the vertices in $P$ have the same value of $y^\star(v)$ for any $v \in P$. Then, it must be that $z^\star = z_P$ and every non-empty level set $Q$ of $z^\star$ equals to $P$, and so the statement holds trivially.

Secondly, we assume that the vertices in $P$ have at least two different $y^\star$-values. We define $\alpha = \min\{y_v^\star : v \in P\}$, and have

$$\alpha \cdot \mathrm{vol}(P) < \sum_{u \in U} d(u) \cdot y_u^\star = 1.$$

We introduce vector $\widehat{z} = (\widehat{x}, \widehat{y})$ defined by

$$\widehat{x}(e) = \begin{cases} \frac{x^\star(e) - \alpha}{1 - \alpha \mathrm{vol}(P)} & \text{if } x^\star(e) \geq 0 \\ 0 & \text{otherwise} \end{cases},$$

and

$$\widehat{y}(v) = \begin{cases} \frac{y^\star(v) - \alpha}{1 - \alpha \mathrm{vol}(P)} & \text{if } v \in P \\ 0 & \text{otherwise} \end{cases},$$

which implies that

$$z^\star = (1 - \alpha \mathrm{vol}(P))\,\widehat{z} + \alpha \cdot \mathbf{1}_P = (1 - \alpha \mathrm{vol}(P))\,\widehat{z} + \alpha \cdot \mathrm{vol}(P) \cdot z_P, \tag{6}$$

where $\mathbf{1}_P$ is the indicator vector of the set $P$. Notice that $\widehat{z}$ preserves the relative ordering of the vertices and edges with respect to $x^\star$ and $y^\star$, and all the constraints in (5) hold for $\widehat{z}$. These imply that $\widehat{z}$ is a feasible solution to (5) as well. Moreover, it's not difficult to see that $\widehat{z}$ is an optimal solution of (5), since otherwise by the linearity of (6), $z_P$ would have a higher objective value than $z^\star$, contradicting the fact that $z^\star$ is an optimal solution. Hence, the non-empty level set defined by $\widehat{z}$ corresponds to an optimal solution.

Finally, by applying the second case inductively, we prove the claimed statement of the lemma. $\square$

By applying the lemma above and the linearity of the objective function of (5), we obtain the following corollary.

**Corollary 1.** *The following statements hold:*

- *Suppose that $P_1$ and $P_2$ are optimal subsets of $U$. Then, $P_1 \cup P_2$, as well as $P_1 \cap P_2$ satisfying $P_1 \cap P_2 \neq \emptyset$, is an optimal subset of $U$.*

- *The optimal set of maximum size is unique, and contains all optimal subsets.*

Now we are ready to show that the procedure `ComputeChangeRate` (Algorithm 2) and the linear program (5) together will always give us the same set of $r$-values regardless of which optimal solution of (5) is used for the recursive construction of the entire vector $r$.

**Lemma 2.** *Let $(U, E_U)$ be the input to `ComputeChangeRate`, and $P \subset U$ be the set returned by (5). Moreover, let $(U' = U \setminus P, E_{U'})$ be the input to the recursive call `ComputeChangeRate`$(U', E_{U'})$. Then, it holds for any $P' \subseteq U'$ that $\delta(P') \leq \delta(P)$, where the equality holds iff $\delta(P \cup P') = \delta(P)$.*

*Proof.* By the definition of the function $c$ and sets $\mathcal{S}^+, \mathcal{S}^-, \mathcal{I}^+, \mathcal{I}^-$, it holds that

$$c\left(\mathcal{S}_{U',P'}^+\right) = c\left(\mathcal{S}_{U,P \cup P'}^+\right) - c\left(\mathcal{S}_{U,P}^+\right),$$

and the same equality holds for sets $\mathcal{S}^-, \mathcal{I}^+$ and $\mathcal{I}^-$. We have that

$$\begin{aligned}
\delta(P') &= \frac{c\left(\mathcal{S}_{U',P'}^+\right) + c\left(\mathcal{I}_{U',P'}^+\right) - c\left(\mathcal{S}_{U',P'}^-\right) - c\left(\mathcal{I}_{U',P'}^-\right)}{\mathrm{vol}(P')} \\
&= \frac{\delta(P \cup P') \cdot \mathrm{vol}(P \cup P') - \delta(P) \cdot \mathrm{vol}(P)}{\mathrm{vol}(P \cup P') - \mathrm{vol}(P)}.
\end{aligned}$$

Therefore, it holds for any operator $\bowtie \in \{<, =, >\}$ that

$$\delta(P') \bowtie \delta(P)$$

$$\iff \quad \frac{\delta(P \cup P') \cdot \mathrm{vol}(P \cup P') - \delta(P) \cdot \mathrm{vol}(P)}{\mathrm{vol}(P \cup P') - \mathrm{vol}(P)} \bowtie \delta(P)$$

$$\iff \quad \delta(P \cup P') \bowtie \delta(P),$$

which implies that $\delta(P') \leq \delta(P)$ iff $\delta(P \cup P') \leq \delta(P)$ with equality iff $\delta(P \cup P') = \delta(P)$. Since $P$ is optimal, it cannot be the case that $\delta(P \cup P') > \delta(P)$, and therefore the lemma follows. $\qquad\square$

Combining everything together, we have the following result which summaries the properties of $r$ computed by Algorithm 2 and (5) and establishes the first fact promised at the beginning of this section.

**Lemma 3.** *For any input instance $(U, E_U)$, Algorithm 2 always returns the same output $r \in \mathbb{R}^{|U|}$ no matter which optimal sets are returned by solving the linear program (5). In particular, Algorithm 2 always finds the unique optimal set $P \subseteq U$ of maximum size and assigns $r(u) = \delta(P)$ to every $u \in P$. After removing the computed $P \subset U$, the computed $r(v) = \delta(P')$ for some $P' \subseteq U \setminus P$ and any $v \in P'$ is always strictly less than $r(u) = \delta(P)$ for any $u \in P$.*

**Any $r$ satisfying Rules (1) and (2) is computed by Algorithm 2.** Next we show that if there is any vector $r$ which satisfies Rules (1) and (2), it must be equal to the output of Algorithm 2.

**Lemma 4.** *For any hypergraph $H = (V_H, E_H, w)$ and $f_t \in \mathbb{R}^n$, if there is a vector $r = \mathrm{d}f_t/\mathrm{d}t$ with an associated $\{r_e(v)\}_{e \in E_H, v \in V_H}$ which satisfies Rule (1) and (2), then $r$ is equal to the output of Algorithm 2.*

*Proof.* We will focus our attention on a single equivalence class $U \subset V$ where for any $u, v \in U$, $f(u) = f(v)$. Recall that for each $e \in E_U$, $c_{f_t}(e) = w(e) |\Delta_{f_t}(e)|$, which is the rate of flow due to $e$ into $U$ (if $e \in S_U^+ \cup I_U^+$) or out of $U$ (if $e \in S_U^- \cup I_U^-$). Let $r \in \mathbb{R}^n$ be the vector supposed to satisfy Rule (1) and (2). We assume that $U \subseteq V$ is an arbitrary equivalence class, and define

$$T \triangleq \left\{ u \in U : r(u) = \max_{v \in U} r(v) \right\}.$$

Let us study which properties $r$ must satisfy according to Rule (1) and (2).

- Assume that $e \in \mathcal{S}_U^-$, i.e., it holds that $S_{f_t}(e) \cap U \neq \emptyset$ and $\Delta_{f_t}(e) > 0$. To satisfy Rule (2a), it suffices to have that $c_{f_t}(e) = w_e \cdot \Delta_{f_t}(e) = -\sum_{v \in S_{f_t}(e)} d(v) r_e(v) = -\sum_{v \in T} d(v) r_e(v)$ if $S_{f_t}(e) \cap T \neq \emptyset$, and $r_e(v) = 0$ for all $v \in T$ otherwise.

- Assume that $e \in \mathcal{S}_U^+$, i.e., it holds that $S_{f_t}(e) \cap U \neq \emptyset$ and $\Delta_{f_t}(e) < 0$. To satisfy Rule (2a), it suffices to have $S_{f_t}(e) \subseteq T$, or $S_{f_t}(e) \cap T = \emptyset$.

- Assume that $e \in \mathcal{I}_U^+$, i.e., it holds that $I_f(e) \cap U \neq \emptyset$ and $\Delta_f(e) < 0$. To satisfy Rule (2b), it suffices to have that $c_f(e) = \sum_{v \in I_f(e)} d(v) r_e(v) = \sum_{v \in T} d(v) r_e(v)$ if $I_f(e) \subseteq T$, and $r_e(v) = 0$ for all $v \in T$ otherwise.

- Assume that $e \in \mathcal{I}_U^-$, i.e., it holds that $I_f(e) \cap U \neq \emptyset$ and $\Delta_f(e) > 0$. To satisfy Rule (2b), it suffices to have $I_f(e) \subseteq T$, or $I_f(e) \cap T = \emptyset$.

Notice that the four conditions above needed to satisfy Rule (2) naturally reflect our definitions of the sets $\mathcal{S}_{U,P}^+, \mathcal{I}_{U,P}^+, \mathcal{S}_{U,P}^-$, and $\mathcal{I}_{U,P}^-$ and for all $u \in T$, it must be that $r(u) = \delta(T)$.

We will show that the output set $P$ returned by solving the linear program (5) is the set $T$. To prove this, notice that on one hand, by Corollary 1, the linear program gives us the unique maximal optimal subset $P \subseteq U$, and every $v \in P$ satisfies that $r(v) = \delta(P) \leq r(u) = \delta(T)$ for any $u \in T$ as every vertex in $T$ has the maximum $r$-value. On the other side, we have that $\delta(T) \leq \delta(P)$ since $P$ is the set returned by the linear program, and therefore $T = P$. We can apply this argument recursively, and this proves that Algorithm 2 must return the vector $r$. $\qquad\square$

**The output of Algorithm 2 satisfies Rules (1) and (2).** Now we show that the output of Algorithm 2 does indeed satisfy Rules (1) and (2) which, together with Lemma 4 implies that there is exactly one such vector which satisfies the rules.

**Lemma 5.** *For any hypergraph $H = (V_H, E_H, w)$ and vector $f_t \in \mathbb{R}^n$, the vector $r$ constructed by Algorithm 2 has corresponding $\{r_e(v)\}_{e \in E_H, v \in V_H}$ which satisfies Rules (1) and (2). Moreover, the $\{r_e(v)\}_{e \in E_H, v \in V_H}$ values can be found in polynomial time using the vector $r$.*

*Proof.* We will focus on a single iteration of the algorithm, in which $r(v)$ is assigned for the vertices in some set $T \subset V_H$. We use the notation

$$\mathcal{E}_T^+ = \mathcal{I}_{U,T}^+ \cup \mathcal{S}_{U,T}^+, \qquad \mathcal{E}_T^- = \mathcal{I}_{U,T}^- \cup \mathcal{S}_{U,T}^-$$

and will show that the values of $r_e(v)$ for $e \in \mathcal{E}_T^+ \cup \mathcal{E}_T^-$ can be found and satisfy Rules (1) and (2). Therefore, by applying this argument to each recursive call of the algorithm, we establish the lemma. Given the set $T$, construct the following undirected flow graph, which is illustrated in Figure 5.

- The vertex set is $\mathcal{E}_T^+ \cup \mathcal{E}_T^- \cup T \cup \{s, t\}$.

- For all $e \in \mathcal{E}_T^+$, there is an edge $(s, e)$ with capacity $c_{f_t}(e)$.

- For all $e \in \mathcal{E}_T^-$, there is an edge $(e, t)$ with capacity $c_{f_t}(e)$.

- For all $v \in T$, if $\delta(T) \geq 0$, there is an edge $(v, t)$ with capacity $d(v)\delta(T)$. Otherwise, there is an edge $(s, v)$ with capacity $d(v) |\delta(T)|$.

- For each $e \in \mathcal{E}_T^+ \cup \mathcal{E}_T^-$, and each $v \in T \cap (S_{f_t}(e) \cup I_{f_t}(e))$, there is an edge $(e, v)$ with capacity $\infty$.

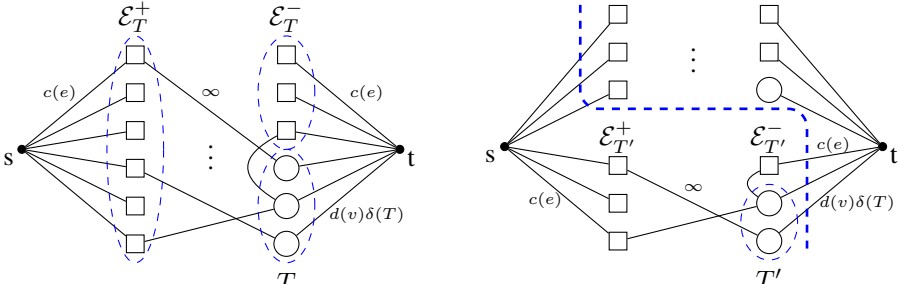

Figure 5: **Left**: an illustration of the constructed max-flow graph, when $\delta(T) \geq 0$. The minimum cut is given by $\{s\}$. **Right**: a cut induced by $T' \subset T$. We can assume that every $e \in \mathcal{E}_T^+ \cup \mathcal{E}_T^-$ connected to $T'$ is on the same side of the cut as $T'$. Otherwise, there would be an edge with infinite capacity crossing the cut.

We use $\mathrm{cut}(A)$ to denote the weight of the cut defined by the set $A$ in this constructed graph and note that $\mathrm{cut}(\{s\}) = \mathrm{cut}(\{t\})$ since

$$\mathrm{cut}(\{s\}) - \mathrm{cut}(\{t\}) = \sum_{e \in \mathcal{E}_T^+} c_f(e) - \sum_{e \in \mathcal{E}_T^-} c_f(e) - \mathrm{vol}(T)\delta(T) = 0$$

by the definition of $\delta(T)$.

Now, suppose that the maximum flow value on this graph is $\mathrm{cut}(\{s\})$, and let the corresponding flow from $u$ to $v$ be given by $\Theta(u, v) = -\Theta(v, u)$. Then, set $d(v)r_e(v) = \Theta(e, v)$ for any $e \in \mathcal{E}_T^+ \cup \mathcal{E}_T^-$ and $v \in T \cap e$. This configuration of the values $r_e(v)$ would be compatible with the vector $r$ computed by Algorithm 2 and would satisfy the rules of the diffusion process for the following reasons:

- For all $v \in T$, the edge $(v, t)$ or $(s, v)$ is saturated and so $\sum_{e \in E} d(v)r_e(v) = d(v)\delta(T) = d(v)r(v)$.

- For any $e \in \mathcal{E}_T^+ \cup \mathcal{E}_T^-$, the edge $(s, e)$ or $(e, t)$ is saturated and so we have that $\sum_{v \in T \cap e} d(v) r_v(e) = -w(e) \Delta(e)$. Since $\mathcal{E}_T^+ \cup \mathcal{E}_T^-$ is removed in the recursive step of the algorithm, $r_e(v) = 0$ for all $v \in U \setminus T$ and so $\sum_{v \in U \cap e} d(v) r_e(v) = -w(e) \Delta(e)$. This establishes Rule (1) since $U \cap e$ is equal to either $S_f(e)$ or $I_f(e)$.

- For edges in $\mathcal{S}_{U,T}^+$ (resp. $\mathcal{I}_{U,T}^+, \mathcal{I}_{U,T}^-$), since $S_{f_t}(e)$ (resp. $I_{f_t}(e), I_{f_t}(e)$) is a subset of $T$ and every $v \in T$ has the same value of $r(v)$, Rule (2) is satisfied. For edges in $\mathcal{S}_{U,T}^-$, for any $v \notin T$, we have $r_e(v) = 0$ and $r(v) < \delta(T)$ which satisfies Rule (2).

We will now show that every cut separating $s$ and $t$ has weight at least $\mathrm{cut}(\{s\})$ which will establish that the maximum flow on this graph is $\mathrm{cut}(\{s\})$ by the max-flow min-cut theorem.

Consider some arbitrary cut given by $X = \{s\} \cup T' \cup \mathcal{E}_{T'}^+ \cup \mathcal{E}_{T'}^-$ where $T'$ (resp. $\mathcal{E}_{T'}^+, \mathcal{E}_{T'}^-$) is a subset of $T$ (resp. $\mathcal{E}_T^+, \mathcal{E}_T^-$). Figure 5 illustrates this cut. Since all of the edges not connected to $s$ or $t$ have infinite capacity, we can assume that no such edge crosses the cut which implies that

- For all $e \in \mathcal{E}_{T'}^+$, $e \cap (T \setminus T') = \emptyset$.

- For all $e \in \mathcal{E}_{T'}^-$, $e \cap (T \setminus T') = \emptyset$.

- For all $e \in (\mathcal{E}_T^+ \setminus \mathcal{E}_{T'}^+)$, $e \cap T' = \emptyset$.

- For all $e \in (\mathcal{E}_T^- \setminus \mathcal{E}_{T'}^-)$, $e \cap T' = \emptyset$.

These conditions, along with the definition of $\mathcal{E}_T^+$ and $\mathcal{E}_T^-$, allow us to assume that $\mathcal{E}_{T'}^+ = \mathcal{I}_{U,T'}^+ \cup \mathcal{S}_{U,T'}^+$ and $\mathcal{E}_{T'}^- = \mathcal{I}_{U,T'}^- \cup \mathcal{S}_{U,T'}^-$. The size of this arbitrary cut is

$$\mathrm{cut}(X) = \mathrm{cut}(\{s\}) - \sum_{e \in \mathcal{E}_{T'}^+} c(e) + \sum_{e \in \mathcal{E}_{T'}^-} c(e) + \sum_{v \in T'} d(v) \delta(T).$$

Since $T$ maximises the objective function $\delta$, we have

$$\sum_{e \in \mathcal{E}_{T'}^+} c(e) - \sum_{e \in \mathcal{E}_{T'}^-} c(e) = \mathrm{vol}(T') \delta(T') \leq \mathrm{vol}(T') \delta(T) = \sum_{v \in T'} d(v) \delta(T)$$

and can conclude that $\mathrm{cut}(X) \geq \mathrm{cut}(\{s\})$ which completes the proof. $\square$

**Proof of Theorem 2.** We can now combine the results in Lemmas 4 and 5 to prove Theorem 2.

*Proof of Theorem 2.* Lemma 4 and Lemma 5 together imply that there is a unique vector $r$ and corresponding $\{r_e(v)\}_{e \in E_H, v \in V_H}$ which satisfies Rules (1) and (2). Lemma 4 further shows that Algorithm 2 computes this vector $r$, and the proof of Lemma 5 gives a polynomial-time method for computing the $\{r_e(v)\}$ values by solving a sequence of max-flow problems. $\square$

# B  Proof of Theorem 1

In this section we will prove Theorem 1. This section is split into two subsections which correspond to the two statements in Theorem 1. First, we show that the diffusion process converges to an eigenvector of $J_H$. We then show that this allows us to find sets $L, R \subset V_H$ with low hypergraph bipartiteness.

## B.1  Convergence of diffusion process

We show in this section that the diffusion process determined by the operator $J_H$ converges in polynomial time to an eigenvector of $J_H$.

**Theorem 3.** *For any $\epsilon > 0$, there is some $t = O\left(1/\epsilon^3\right)$ such that for any starting vector $f_0$, there is an interval $[c, c + 2\epsilon]$ such that*

$$\frac{1}{\|f_t\|_w} \sum_{u_i : \lambda_i \in [c, c+2\epsilon]} \langle f_t, u_i \rangle_w \geq 1 - \epsilon$$

*where $(u_i, \lambda_i)$ are the eigen-pairs of $D_H^{-1} J_t = D_H^{-1}(D_{G_t} + A_{G_t})$ and $G_t$ is the graph constructed by the diffusion operator $J_H$ at time $t$.*

By taking $\epsilon$ to be some small constant, this shows that the vector $f_t$ converges to an eigenvector of the hypergraph operator in polynomial time.

*Proof.* We will prove this by showing that the Rayleigh quotient $R_H(f_t)$ is always decreasing at a rate of at least $\epsilon^3$ whenever the conclusion of Theorem 3 does not hold. Since $R_H(f_t)$ can only decrease by a constant amount, the theorem will follow.

First, we derive an expression for the rate of change of the Rayleigh quotient $R_H(f_t)$. Let $x_t \triangleq D_H^{1/2} f_t$, and let $\mathcal{J}_t = D_H^{-1/2}(D_{G_t} + A_{G_t})D_H^{-1/2}$. Then, we have

$$R_H(f_t) = \frac{f_t^\mathsf{T}(D_{G_t} + A_{G_t})f_t}{f_t^\mathsf{T} D_H f_t} = \frac{x_t^\mathsf{T} \mathcal{J}_t x_t}{x_t^\mathsf{T} x_t}.$$

For every eigenvector $u_j$ of $D_H^{-1} J_H$, the vector $v_j = D_H^{1/2} u_j$ is an eigenvector of $\mathcal{J}$ with the same eigenvalue $\lambda_i$. Since $\mathcal{J}$ is symmetric, the vectors $v_1$ to $v_n$ are orthogonal. Additionally, notice that

$$\langle x_t, v_j \rangle = f_t^\mathsf{T} D_H u_j = \langle f_t, u_j \rangle_w$$

and we define $\alpha_j = \langle f_t, u_j \rangle_w$ so we can write $x_t = \sum_{j=1}^n \alpha_j v_j$ and $f_t = D_H^{-1/2} x_t = \sum_{j=1}^n \alpha_j u_j$. Now, we have that

$$\frac{\mathrm{d}}{\mathrm{d}t}\langle x_t, \mathcal{J}_t x_t \rangle = \langle \frac{\mathrm{d}}{\mathrm{d}t} x_t, \mathcal{J}_t x_t \rangle + \langle x_t, \frac{\mathrm{d}}{\mathrm{d}t}\mathcal{J}_t x_t \rangle$$
$$= -x_t^\mathsf{T} \mathcal{J}_t^2 x_t - x_t^\mathsf{T} \mathcal{J}_t^2 x_t$$
$$= -2\sum_{j=1}^n \alpha_j^2 \lambda_j^2.$$

Additionally, we have

$$\frac{\mathrm{d}}{\mathrm{d}t} x_t^\mathsf{T} x_t = -x_t^\mathsf{T} \mathcal{J}_t x_t - x_t^\mathsf{T} \mathcal{J}_t x_t = -2(x_t^\mathsf{T} \mathcal{J}_t x_t).$$

Recalling that $x_t^\mathsf{T} x_t = \sum_{j=1}^n \alpha_j^2$, this gives

$$\frac{d}{dt}R(f_t) = \frac{1}{(x_t^\mathsf{T} x_t)^2}\left[\left(\frac{\mathrm{d}}{\mathrm{d}t}(x_t^\mathsf{T} \mathcal{J}_t x_t)\right)(x_t^\mathsf{T} x_t) - \left(\frac{\mathrm{d}}{\mathrm{d}t}(x_t^\mathsf{T} x_t)\right)(x_t^\mathsf{T} \mathcal{J}_t x_t)\right]$$
$$= \frac{1}{x_t^\mathsf{T} x_t}\left(\frac{\mathrm{d}}{\mathrm{d}t}(x_t^\mathsf{T} \mathcal{J}_t x_t)\right) + 2R(f_t)^2$$
$$= 2\left[R(f_t)^2 - \frac{1}{\sum_{j=1}^n \alpha_j^2}\sum_{j=1}^n \alpha_j^2 \lambda_j^2\right] \tag{7}$$

We will now show that at any time $t$, if the conclusion of the theorem does not hold, then

$$\frac{\mathrm{d}}{\mathrm{d}t}R(f_t) \leq -\epsilon^3. \tag{8}$$

Assuming that this holds, and using the fact that for all $t$ it is the case that $0 \leq R(f_t) \leq 2$, when $t = 2/\epsilon^3$, either $R(f_t) = 0$ or there was some $t' < t$ when $(\mathrm{d}/\mathrm{d}t')R(f_{t'}) > -\epsilon^3$ and the conclusion of the theorem holds.

Now, to show (8), consider the partial derivative

$$\frac{\partial}{\partial \lambda_i} 2\left[R(f_t)^2 - \frac{1}{\sum_{j=1}^n \alpha_j^2}\sum_{j=1}^n \alpha_j^2 \lambda_j^2\right] = 2\left[\frac{2\alpha_i^2}{\sum_{j=1}^n \alpha_j^2}R(f_t) - \frac{2\alpha_i^2}{\sum_{j=1}^n \alpha_j^2}\lambda_i\right]$$

$$= \frac{4\alpha_i^2}{\sum_{j=1}^n \alpha_j^2}(R(f_t) - \lambda_i), \tag{9}$$

where we use the fact that $R(f_t) = (\sum_{j=1}^n \alpha_j^2 \lambda_j)/(\sum_{j=1}^n \alpha_j^2)$. Notice that if $\lambda_i < R(f_t)$, the derivative in (9) is greater than 0 and if $\lambda_i > R(f_t)$, the derivative is less than 0. This means that in order to establish an upper-bound for $(\mathrm{d}/\mathrm{d}t)R(f_t)$, we can assume that the eigenvalues $\lambda_j$ are as close to the value of $R(f_t)$ as possible.

Now, we assume that at time $t$, the conclusion of the theorem does not hold. Then, one of the following cases must hold:

1. $(\sum_{j:\lambda_j > R(f_t)+\epsilon} \alpha_j^2)/(\sum_{j=1}^n \alpha_j^2) > \epsilon/2$

2. $(\sum_{j:\lambda_j < R(f_t)-\epsilon} \alpha_j^2)/(\sum_{j=1}^n \alpha_j^2) > \epsilon/2$

Suppose the first case holds. By the conclusions we draw from (9), we can assume that there is an eigenvalue $\lambda_i = R(f_t)+\epsilon$ such that $\alpha_i^2/(\sum_{j=1}^n \alpha_j^2) = \epsilon/2$ and that $(\sum_{j:\lambda_j < R(f_t)} \alpha_j^2)/(\sum_{j=1}^n \alpha_j^2) = 1 - \epsilon/2$. Then, since $R(f_t) = (\sum_{j=1}^n \alpha_j^2 \lambda_j)/(\sum_{j=1}^n \alpha_j^2)$, we have

$$\frac{1}{\sum_{j=1}^n \alpha_j^2}\sum_{j:\lambda_j < R(f_t)} \alpha_j^2 \lambda_j = R(f_t) - \frac{\epsilon}{2}(R(f_t) + \epsilon),$$

which is equivalent to

$$\frac{1}{\sum_{j=1}^n \alpha_j^2}\sum_{j:\lambda_j < R(f_t)} \alpha_j^2(\lambda_j - R(f_t))) = \frac{\epsilon}{2}\cdot R(f_t) - \frac{\epsilon}{2}\cdot(R(f_t) + \epsilon) = -\frac{\epsilon^2}{2}. \tag{10}$$

Now, notice that for any $\lambda_j < R(f_t))$ we have

$$(\lambda_j^2 - R(f_t)^2) = (\lambda_j + R(f_t))(\lambda_j - R(f_t))$$
$$\geq 2R(f_t)(\lambda_j - R(f_t)),$$

since $\lambda_j - R(f_t) < 0$. As such, we have

$$\frac{\mathrm{d}}{\mathrm{d}t}R(f_t) = 2\left[R(f_t)^2 - \frac{1}{\sum_{j=1}^n \alpha_j^2}\sum_{j=1}^n \alpha_j^2 \lambda_j^2\right]$$

$$= -\frac{2}{\sum_{j=1}^n \alpha_j^2}\sum_{j=1}^n \alpha_j^2(\lambda_j^2 - R(f_t)^2)$$

$$= -\epsilon\left((R(f_t) + \epsilon)^2 - R(f_t)^2\right) - \frac{2}{\sum_{j=1}^n \alpha_j^2}\sum_{j:\lambda_j < R(f_t)} \alpha_j^2(\lambda_j^2 - R(f_t)^2)$$

$$\leq -\epsilon\left(2\epsilon R(f_t) + \epsilon^2\right) - \frac{2}{\sum_{j=1}^n \alpha_j^2}\sum_{j:\lambda_j < R(f_t)} 2\alpha_j^2 R(f_t)(\lambda_j - R(f_t))$$

$$= -2\epsilon^2 R(f_t) - \epsilon^3 + 2\epsilon^2 R(f_t)$$

$$= -\epsilon^3$$

where the fifth line follows by (10). We now consider the second case. We can assume that there is an eigenvalue $\lambda_i = R(f_t)-\epsilon$ such that $\alpha_i^2/(\sum_{j=1}^n \alpha_j^2) = \epsilon/2$ and that $(\sum_{j:\lambda_j > R(f_t)} \alpha_j^2)/(\sum_{j=1}^n \alpha_j^2) = 1 - \epsilon/2$. Then, we have

$$\frac{1}{\sum_{j=1}^n \alpha_j^2}\sum_{j:\lambda_j > R(f_t)} \alpha_j^2 \lambda_j = R(f_t) - \frac{\epsilon}{2}(R(f_t) - \epsilon),$$

which is equivalent to

$$\frac{1}{\sum_{j=1}^n \alpha_j^2} \sum_{j:\lambda_j > R(f_t)} \alpha_j^2(\lambda_j - R(f_t)) = \frac{\epsilon}{2}R(f_t) - \frac{\epsilon}{2}(R(f_t) - \epsilon) = \frac{\epsilon^2}{2}. \tag{11}$$

Now, notice that for any $\lambda_j > R(f_t)$ we have

$$\lambda_j^2 - R(f_t)^2 = (\lambda_j + R(f_t))(\lambda_j - R(f_t))$$
$$\geq 2R(f_t) \cdot (\lambda_j - R(f_t)).$$

As such, we have

$$\frac{\mathrm{d}}{\mathrm{d}t}R(f_t) = -\frac{2}{\sum_{j=1}^n \alpha_j^2} \sum_{j=1}^n \alpha_j^2(\lambda_j^2 - R(f_t)^2)$$

$$= -\epsilon\left((R(f_t) - \epsilon)^2 - R(f_t)^2\right) - \frac{2}{\sum_{j=1}^n \alpha_j^2} \sum_{j:\lambda_j > R(f_t)} \alpha_j^2(\lambda_j^2 - R(f_t)^2)$$

$$\leq -\epsilon\left(\epsilon^2 - 2\epsilon R(f_t)\right) - \frac{2}{\sum_{j=1}^n \alpha_j^2} \sum_{j:\lambda_j < R(f_t)} 2\alpha_j^2 R(f_t)(\lambda_j - R(f_t))$$

$$= 2\epsilon^2 g(t) - \epsilon^3 - 2\epsilon^2 R(f_t)$$

$$= -\epsilon^3$$

where the fourth line follows by (11). These two cases establish (8) and complete the proof of the theorem. $\square$

**The eigenvector is at most the minimum eigenvector of the clique graph.** We now show that the eigenvector to which we converge is at most the minimum eigenvector of $J_G$ where $G$ is the clique reduction of $H$. We start by showing the following technical lemma.

**Lemma 6.** *For any hypergraph $H = (V_H, E_H, w)$, vector $f \in \mathbb{R}^n$, and edge $e \in E_H$, it is the case that*

$$\left(\max_{u \in e} f(u) + \min_{v \in e} f(v)\right)^2 \leq \sum_{u,v \in e} \frac{1}{r_e - 1}(f(u) + f(v))^2$$

*where $r_e$ is the rank of the edge $e$. The equality holds iff there is exactly one vertex $v \in e$ with $f(v) \neq 0$ or $r_e = 2$.*

*Proof.* We will consider some ordering of the vertices in $e$,

$$u_1, u_2, \ldots, u_{r_e},$$

such that $u_1 = \arg\max_{u \in e} f(u)$ and $u_2 = \arg\min_{u \in e} f(u)$ and the remaining vertices are ordered arbitrarily. Then, for any $2 \leq k \leq r_e$, we define

$$C_k = \sum_{u,v \in \{u_1, \ldots, u_k\}} \frac{1}{k-1}(f(u) + f(v))^2$$

and we will show by induction on $k$ that

$$C_k \geq \left(\max_{u \in e} f(u) + \min_{v \in e} f(v)\right)^2 \tag{12}$$

for all $2 \leq k \leq r_e$ with equality iff $k = 2$ or there is exactly one vertex $u_i \in e$ with $f(v) \neq 0$. The lemma follows by setting $k = r_e$.

The base case when $k = 2$ follows trivially by the definitions and the choice of $u_1$ and $u_2$.

For the inductive step, we assume that (12) holds for some $k$ and will show that it holds for $k + 1$. We have that

$$C_{k+1} = \sum_{u,v \in \{u_1, \ldots, u_{k+1}\}} \frac{1}{k} \cdot (f(u) + f(v))^2$$

which is equivalent to

$$C_{k+1} = \frac{1}{k} \sum_{i=1}^{k} (f(u_i) + f(u_{k+1}))^2 + \frac{1}{k} \sum_{u,v \in \{u_1,\dots,u_k\}} (f(u) + f(v))^2$$

$$= \frac{1}{k} \sum_{i=1}^{k} (f(u_i) + f(u_{k+1}))^2 + \frac{k-1}{k} C_k$$

$$\geq \left(1 - \frac{1}{k}\right) \left(\max_{u \in e} f(u) + \min_{v \in e} f(v)\right)^2 + \frac{1}{k} \sum_{i=1}^{k} (f(u_1) + f(u_{k+1}))^2$$

where the final inequality holds by the induction hypothesis. Therefore, it is sufficient to show that

$$\sum_{i=1}^{k} (f(u_i) + f(u_{k+1}))^2 \geq \left(\max_{u \in e} f(u) + \min_{v \in e} f(v)\right)^2 .$$

We will instead show the stronger fact that

$$\left(\max_{v \in e} f(v) + f(u_{k+1})\right)^2 + \left(\min_{v \in e} f(v) + f(u_{k+1})\right)^2 \geq \left(\max_{u \in e} f(u) + \min_{v \in e} f(v)\right)^2 . \tag{13}$$

The proof is by case distinction. The first case is when $\text{sign}(\max_{u \in e} f(v)) = \text{sign}(\min_{u \in e} f(u))$. Assume w.l.o.g. that the sign is positive. Then, since $f(u_{k+1}) \geq \min_{v \in e} f(v)$, we have

$$\left(\max_{v \in e} f(v) + f(u_{k+1})\right)^2 \geq \left(\max_{v \in e} f(v) + \min_{u \in e} f(v)\right)^2$$

and (13) holds. Moreover, the inequality is strict if $|\min_{u \in e} f(u)| > 0$ or $|f(u_{k+1})| > 0$.

For the second case, we assume that $\text{sign}(\min_{u \in e} f(u)) \neq \text{sign}(\max_{v \in e} f(v))$. Expanding (13), we would like to show

$$\left(\max_{u \in e} f(u)\right)^2 + \left(\min_{v \in e} f(v)\right)^2 + 2f(u_{k+1})\left(\max_{u \in e} f(u)\right) + 2f(u_{k+1})\left(\min_{v \in e} f(v)\right) + 2f(u_{k+1})^2$$

$$\geq \left(\max_{u \in e} f(u)\right)^2 + \left(\min_{u \in e} f(u)\right)^2 - 2\left(\max_{u \in e} f(u)\right)\left|\min_{u \in e} f(u)\right|$$

which is equivalent to

$$2f(u_{k+1})^2 + 2f(u_{k+1})\left(\max_{u \in e} f(u)\right) + 2f(u_{k+1})\left(\min_{v \in e} f(v)\right) \geq -2\left(\max_{u \in e} f(u)\right)\left|\min_{v \in e} f(v)\right| .$$

Notice that exactly one of the terms on the left hand side is negative. Recalling that $\min_{u \in e} f(u) \leq f(u_{k+1}) \leq \max_{v \in e} f(v)$, it is clear that

- if $f(u_{k+1}) < 0$, then $-2\left(\max_{u \in e} f(u)\right)\left|\min_{v \in e} f(v)\right| \leq 2f(u_{k+1})\left(\max_{v \in e} f(v)\right) \leq 0$ and the inequality holds.

- if $f(u_{k+1}) \geq 0$, then $-2\left(\max_{u \in e} f(u)\right)\left|\min_{v \in e} f(v)\right| \leq 2f(u_{k+1})\left(\min_{u \in e} f(u)\right) \leq 0$ and the inequality holds.

Moreover, in both cases the inequality is strict if $-2\left(\max_{v \in e} f(v)\right)\left|\min_{u \in e} f(u)\right| < 0$ or $|f(u_{k_1})| > 0$. $\qquad \square$

Now, we can show that we always find an eigenvector which is at most the minimum eigenvector of the clique reduction.

**Lemma 7.** *For any hypergraph* $H = (V_H, E_H, w)$ *with clique reduction* $G$, *if* $f$ *is the eigenvector corresponding to* $\lambda_1(D_G^{-1} J_G)$, *then*

$$\frac{f^\intercal J_H f}{f^\intercal D_H f} \leq \lambda_1(D_G^{-1} J_G)$$

*and the inequality is strict if* $\min_{e \in E_H} r_e > 2$.

*Proof.* Since $\lambda_1(D_G^{-1}J_G) = (f^\mathsf{T} J_G f)/(f^\mathsf{T} D_G f)$ and $(f^\mathsf{T} D_G f) = (f^\mathsf{T} D_H f)$ by the construction of the clique graph, it suffices to show that

$$f^\mathsf{T} J_H f \le f^\mathsf{T} J_G f$$

which is equivalent to

$$\sum_{e \in E_H} w(e) \left( \max_{v \in e} f(v) + \min_{u \in e} f(u) \right)^2 \le \sum_{(u,v) \in E_G} w_G(u,v)(f(u) + f(v))^2$$

$$= \sum_{e \in E_H} w(e) \sum_{u,v \in e} \frac{1}{r_e - 1}(f(u) + f(v))^2$$

which holds by Lemma 6.

Furthermore, if $\min_{e \in E_H} r_e > 2$, then by Lemma 6 the inequality is strict unless every edge $e \in E_H$ contains at most one $v \in e$ with $f(v) \ne 0$. Suppose the inequality is not strict, then it must be that

$$\lambda_1(D_G^{-1}J_G) = \frac{\sum_{(u,v) \in E_G} w_G(u,v)(f(v) + f(u))^2}{\sum_{v \in V_G} d_G(v)f(v)^2}$$

$$= \frac{\sum_{v \in V_G} d_G(v)f(v)^2}{\sum_{v \in V_G} d_G(v)f(v)^2}$$

$$= 1$$

since for every edge $(u,v) \in E_G$, at most one of $f(u)$ or $f(v)$ is not equal to 0. This cannot be the case, since it is a standard fact that the maximum eigenvalue $\lambda_n(D_G^{-1}J_G) = 2$ and so $\sum_{i=1}^n \lambda_i(D_G^{-1}J_G) \ge (n-1) + 2 = n + 1$ which contradicts the fact that the trace $\mathrm{tr}(D_G^{-1}J_G)$ is equal to $n$. This proves the final statement of the lemma. $\qquad\square$

### B.2 Cheeger-type inequality for hypergraph bipartiteness

**The operator $J_H$ has a well-defined minimum eigenvector.** Before proving Theorem 2, we will prove some intermediate facts about the new hypergraph operator $J_H$ which will allow us to show that the operator $J_H$ has a well-defined minimum eigenvector. Given a hypergraph $H = (V_H, E_H, w)$, for any edge $e \in E_H$ and weighted measure vector $f_t$, let $r_e^S \triangleq \max_{v \in S_{f_t}(e)}\{r(v)\}$ and $r_e^I \triangleq \min_{v \in I_{f_t}(e)}\{r(v)\}$, and recall that $c_{f_t}(e) = w(e)\,|\Delta_{f_t}(e)|$.

**Lemma 8.** *Given a hypergraph $H = (V_H, E_H, w)$ and normalised measure vector $f_t$, let*

$$r = \frac{\mathrm{d}f_t}{\mathrm{d}t} = -D_H^{-1}J_H f_t.$$

*Then, it holds that*

$$\|r\|_w^2 = -\sum_{e \in E} w(e)\Delta_{f_t}(e)\left(r_e^S + r_e^I\right).$$

*Proof.* Let $P \subset V$ be one of the densest vertex sets defined with respect to the solution of the linear program (5). By the description of Algorithm 2, we have $r(u) = \delta(P)$ for every $u \in P$, and therefore

$$\sum_{u \in P} d(u)r(u)^2 = \mathrm{vol}(P) \cdot \delta(P)^2$$

$$= \left( c_{f_t}\left(\mathcal{S}_{U,P}^+\right) + c_{f_t}\left(\mathcal{I}_{U,P}^+\right) - c_{f_t}\left(\mathcal{S}_{U,P}^-\right) - c_{f_t}\left(\mathcal{I}_{U,P}^-\right) \right) \cdot \delta(P)$$

$$= \left( \sum_{e \in \mathcal{S}_{U,P}^+} c_{f_t}(e) + \sum_{e \in \mathcal{I}_{U,P}^+} c_{f_t}(e) - \sum_{e \in \mathcal{S}_{U,P}^-} c_{f_t}(e) - \sum_{e \in \mathcal{I}_{U,P}^-} c_{f_t}(e) \right) \cdot \delta(P)$$

$$= \sum_{e \in \mathcal{S}_{U,P}^+} c_{f_t}(e) \cdot r_e^S + \sum_{e \in \mathcal{I}_{U,P}^+} c_{f_t}(e) \cdot r_e^I - \sum_{e \in \mathcal{S}_{U,P}^-} c_{f_t}(e) \cdot r_e^S - \sum_{e \in \mathcal{I}_{U,P}^-} c_{f_t}(e) \cdot r_e^I.$$

Since each vertex is included in exactly one set $P$ and each edge will appear either in one each of $\mathcal{S}_{U,P}^+$ and $\mathcal{I}_{U,P}^+$ or in one each of $\mathcal{S}_{U,P}^-$ and $\mathcal{I}_{U,P}^-$, it holds that

$$
\begin{aligned}
\|r\|_w^2 &= \sum_{v \in V} d(v) r(v)^2 \\
&= \sum_P \sum_{v \in P} d(v) r(v)^2 \\
&= \sum_P \left( \sum_{e \in \mathcal{S}_{U,P}^+} c_{f_t}(e) \cdot r_e^S + \sum_{e \in \mathcal{I}_{U,P}^+} c_{f_t}(e) \cdot r_e^I - \sum_{e \in \mathcal{S}_{U,P}^-} c_{f_t}(e) \cdot r_e^S - \sum_{e \in \mathcal{I}_{U,P}^-} c_{f_t}(e) \cdot r_e^I \right) \\
&= - \sum_{e \in E} w(e) \Delta_{f_t}(e) \left( r_e^S + r_e^I \right),
\end{aligned}
$$

which proves the lemma. $\qquad \square$

Next, we define $\gamma_1 = \min_f D(f)$ and now show that any vector $f$ that satisfies $\gamma_1 = D(f)$ is an eigenvector of $J_H$ with eigenvalue $\gamma_1$. We start by showing that the Raleigh quotient of the new operator is equivalent to the discrepancy ratio of the hypergraph.

**Lemma 9.** *For any hypergraph $H$ and vector $f_t \in \mathbb{R}^n$, it holds that $D(f_t) = R_H(f_t)$.*

*Proof.* Since $f_t^{\mathsf{T}} D_H f_t = \sum_{v \in V} d(v) f_t(v)^2$, it is sufficient to show that

$$
f_t^{\mathsf{T}} J_H f_t = \sum_{e \in E_H} w(e) \left( \max_{u \in e} f_t(u) + \min_{v \in e} f_t(v) \right)^2
$$

Recall that for some graph $G_t$, $J_H = D_{G_t} + A_{G_t}$. Then, we have that

$$
\begin{aligned}
f_t^{\mathsf{T}} J_H f_t &= f_t^{\mathsf{T}} (D_{G_t} + A_{G_t}) f_t \\
&= \sum_{(u,v) \in E_G} w_G(u,v)(f_t(u) + f_t(v))^2 \\
&= \sum_{e \in E_H} \sum_{(u,v) \in S_{f_t}(e) \times I_{f_t}(e)} w_{G_t}(u,v)(f_t(u) + f_t(v))^2 \\
&= \sum_{e \in E_H} w(e) \left( \max_{u \in e} f_t(u) + \min_{v \in e} f_t(v) \right)^2,
\end{aligned}
$$

which follows since the graph $G_t$ is constructed by splitting the weight of each hyperedge $e \in E_H$ between the edges $S_{f_t}(e) \times I_{f_t}(e)$. $\qquad \square$

**Lemma 10.** *For a hypergraph $H$, operator $J_H$, and vector $f_t$, the following statements hold:*

1. $\frac{\mathrm{d}}{\mathrm{d}t} \|f_t\|_w^2 = -2 f_t^{\mathsf{T}} J_H f_t$;

2. $\frac{\mathrm{d}}{\mathrm{d}t} (f_t^{\mathsf{T}} J_H f_t) = -2 \|D_H^{-1} J_H f_t\|_w^2$;

3. $\frac{\mathrm{d}}{\mathrm{d}t} R(f_t) \leq 0$ *with equality if and only if* $D_H^{-1} J_H f_t \in \mathrm{span}(f_t)$.

*Proof.* By definition, we have that

$$\frac{\mathrm{d}\|f_t\|_w^2}{\mathrm{d}t} = \frac{\mathrm{d}}{\mathrm{d}t} \sum_{v \in V} d(v) f_t(v)^2$$

$$= \sum_{v \in V} d(v) \cdot \frac{\mathrm{d} f_t(v)^2}{\mathrm{d}t}$$

$$= \sum_{v \in V} d(v) \cdot \frac{\mathrm{d} f_t(v)^2}{\mathrm{d} f_t(v)} \frac{\mathrm{d} f_t(v)}{\mathrm{d}t}$$

$$= 2 \sum_{v \in V} d(v) f_t(v) \cdot \frac{\mathrm{d} f_t(v)}{\mathrm{d}t}$$

$$= 2 \left\langle f_t, \frac{\mathrm{d} f_t}{\mathrm{d}t} \right\rangle_w = -2\langle f_t, D_H^{-1} J_H f_t \rangle_w,$$

which proves the first statement.

For the second statement, by Lemma 9 we have

$$f_t^\mathsf{T} J_H f_t = \sum_{e \in E} w(e) \left( \max_{u \in e} f_t(u) + \min_{v \in e} f_t(v) \right)^2,$$

and therefore

$$\frac{\mathrm{d}}{\mathrm{d}t} f_t^\mathsf{T} J_H f_t = \frac{\mathrm{d}}{\mathrm{d}t} \sum_{e \in E} w(e) \left( \max_{u \in e} f_t(u) + \min_{v \in e} f_t(v) \right)^2$$

$$= 2 \sum_{e \in E} w(e) \Delta_{f_t}(e) \cdot \frac{\mathrm{d}}{\mathrm{d}t} \left( \max_{u \in e} f_t(u) + \min_{v \in e} f_t(v) \right)$$

$$= 2 \sum_{e \in E} w(e) \Delta_{f_t}(e) \cdot \left( r_e^S + r_e^I \right), \tag{14}$$

where the last equality holds by the way that all the vertices receive their $r$-values by the algorithm and the definitions of $r_e^S$ and $r_e^I$. On the other side, by definition $r = -D_H^{-1} J_H f_t$ and so by Lemma 8,

$$\|D_H^{-1} J_H f_t\|_w^2 = \|r\|_w^2 = -\sum_{e \in E} w(e) \Delta_{f_t}(e) \left( r_e^S + r_e^I \right). \tag{15}$$

By combining (14) with (15), we have the second statement.

For the third statement, notice that we can write $f_t^\mathsf{T} J_H f_t$ as $\langle f_t, D_H^{-1} J_H f_t \rangle_w$. Then, we have that

$$\frac{\mathrm{d}}{\mathrm{d}t} \frac{\langle f_t, D_H^{-1} J_H f_t \rangle_w}{\|f_t\|_w^2} = \frac{1}{\|f_t\|_w^2} \cdot \frac{\mathrm{d} \langle f_t, D_H^{-1} J_H f_t \rangle_w}{\mathrm{d}t} - \langle f_t, D_H^{-1} J_H f_t \rangle_w \cdot \frac{1}{\|f_t\|_w^4} \frac{\mathrm{d} \|f_t\|_w^2}{\mathrm{d}t}$$

$$= -\frac{1}{\|f_t\|_w^4} \cdot \left( 2 \cdot \|f_t\|_w^2 \cdot \|D_H^{-1} J_H f_t\|_w^2 + \langle f_t, D_H^{-1} J_H f_t \rangle_w \cdot \frac{\mathrm{d} \|f_t\|_w^2}{\mathrm{d}t} \right)$$

$$= -\frac{2}{\|f_t\|_w^4} \cdot \left( \|f_t\|_w^2 \cdot \|D_H^{-1} J_H f_t\|_w^2 - \langle f_t, D_H^{-1} J_H f_t \rangle_w^2 \right)$$

$$\leq 0,$$

where the last inequality holds by the Cauchy-Schwarz inequality on the inner product $\langle \cdot, \cdot \rangle_w$ with the equality if and only if $D_H^{-1} J_H f_t \in \mathrm{span}(f_t)$. $\qquad \square$

This allows us to establish the following key fact.

**Lemma 11.** *For any hypergraph $H$, $\gamma_1 = \min_f D(f)$ is an eigenvalue of $D_H^{-1} J_H$ and any minimiser $f$ is its corresponding eigenvector.*

*Proof.* By Lemma 9, it holds that $R(f) = D(f)$ for any $f \in \mathbb{R}^n$. When $f$ is a minimiser of $D(f)$, it must hold that

$$\frac{\mathrm{d}R(f)}{\mathrm{d}t} = 0,$$

which implies by Lemma 10 that $D_H^{-1} J_H f \in \mathrm{span}(f)$ and proves that $f$ is an eigenvector. $\quad\square$

**Cheeger-type inequality.** We are now able to prove a Cheeger-type inequality for our operator and the hypergraph bipartiteness.

**Lemma 12.** *Given a hypergraph $H = (V_H, E_H)$ with sets $L, R \subset V_H$ such that $\beta(L, R) = \beta$, it is the case that*

$$\gamma_1 \le 2\beta$$

*where $\gamma_1$ is the smallest eigenvalue of $D_H^{-1} J_H$.*

*Proof.* Let $\chi_{L,R} \in \{-1, 0, 1\}^n$ be the indicator vector of the cut $L, R$ such that

$$\chi_{L,R}(u) = \begin{cases} 1 & \text{if } u \in L \\ -1 & \text{if } u \in R \\ 0 & \text{otherwise} \end{cases}.$$

Then, by Lemma 9, the Rayleigh quotient is given by

$$\begin{aligned}
R_H(\chi_{L,R}) &= \frac{\sum_{e \in E} w(e) \left(\max_{u \in e} \chi_{L,R}(u) + \min_{v \in e} \chi_{L,R}(v)\right)^2}{\sum_{v \in V} d(v) \chi_{L,R}(v)^2} \\
&= \frac{4w(L|\overline{L}) + 4w(R|\overline{R}) + w(L, \overline{L \cup R}|R) + w(R, \overline{L \cup R}|L)}{\mathrm{vol}(L \cup R)} \\
&\le 2\beta(L, R),
\end{aligned}$$

which completes the proof since $\gamma_1 = \min_f R_H(f)$ by Lemma 11. $\quad\square$

The proof of the other direction of the Cheeger-type inequality is more involved, and forms the basis of the second claim in Theorem 1.

**Theorem 4.** *For any hypergraph $H = (V, E, w)$, let $J_H$ be the operator defined with respect to $H$, and $\gamma_1 = \min_f D(f) = \min_f R(f)$ be the minimum eigenvalue of $D_H^{-1} J_H$. Then, there are disjoint sets $L, R \subset V$ such that*

$$\beta(L, R) \le \sqrt{2\gamma_1}.$$

*Proof.* Let $f \in \mathbb{R}^n$ be the vector such that $D(f) = \gamma_1$. For any threshold $t \in [0, \max_u f(u)^2]$, define $X_t$ such that

$$X_t(u) = \begin{cases} 1 & \text{if } f(u) \ge \sqrt{t} \\ -1 & \text{if } f(u) \le -\sqrt{t} \\ 0 & \text{otherwise} \end{cases}.$$

We will show that choosing $t \in \left[0, \max_u f(u)^2\right]$ uniformly at random gives

$$\mathbb{E}\left[\sum_{e \in E_H} w(e) \left|\max_{u \in e} X_t(u) + \min_{v \in e} X_t(v)\right|\right] \le \sqrt{2\gamma_1} \cdot \mathbb{E}\left[\sum_{v \in V_H} d(v) |X_t(v)|\right]. \quad (16)$$

Notice that every such $X_t$ defines disjoints vertex sets $L$ and $R$ that satisfies

$$\beta(L, R) = \frac{\sum_{e \in E_H} w(e) \left|\max_{u \in e} X_t(u) + \min_{v \in e} X_t(v)\right|}{\sum_{v \in V_H} d(v) |X_t(v)|}.$$

Hence, (16) would imply that there is some $X_t$ such that the disjoint $L, R$ defined by $X_t$ would satisfy

$$\sum_{e \in E_H} w(e) \left|\max_{u \in e} X_t(u) + \min_{v \in e} X_t(v)\right| \le \sqrt{2\gamma_1} \cdot \sum_{v \in V_H} d(v) |X_t(v)|,$$

which implies that
$$\beta(L, R) \leq \sqrt{2\gamma_1}.$$
Hence, it suffices to prove (16). We assume without loss of generality that $\max_u \{f(u)^2\} = 1$, so $t$ is chosen uniformly from $[0, 1]$. First of all, we have that
$$\mathbb{E}\left[\sum_{v \in V_H} d(v) |X_t(v)|\right] = \sum_{v \in V_H} d(v)\mathbb{E}\left[|X_t(v)|\right] = \sum_{v \in V_H} d(v)f(v)^2.$$

To analyse the left-hand side of (16), we will focus on a particular edge $e \in E_H$. Let $x_e = \max_{u \in e} f(u)$ and $y_e = \min_{v \in e} f(v)$. We will drop the subscript $e$ when it is clear from context. We will show that
$$\mathbb{E}\left[\left|\max_{u \in e} X_t(u) + \min_{v \in e} X_t(v)\right|\right] \leq |x_e + y_e| (|x_e| + |y_e|). \tag{17}$$

1. Suppose $\text{sign}(x_e) = \text{sign}(y_e)$. Our analysis is by case distinction:
   - $|\max_{u \in e} X_t(u) + \min_{v \in e} X_t(v)| = 2$ with probability $\min(x_e^2, y_e^2)$;
   - $|\max_{u \in e} X_t(u) + \min_{v \in e} X_t(v)| = 1$ with probability $|x_e^2 - y_e^2|$;
   - $|\max_{u \in e} X_t(u) + \min_{v \in e} X_t(v)| = 0$ with probability $1 - \max(x_e^2, y_e^2)$.

   Assume without loss of generality that $x_e^2 = \min(x_e^2, y_e^2)$. Then, it holds hat
   $$\mathbb{E}\left[\left|\max_{u \in e} X(u) + \min_{v \in e} X(v)\right|\right] = 2x_e^2 + |x_e^2 - y_e^2| = x_e^2 + y_e^2 \leq |x_e + y_e| (|x_e| + |y_e|).$$

2. Suppose $\text{sign}(x_e) \neq \text{sign}(y_e)$. Our analysis is by case distinction:
   - $|\max_{u \in e} X_t(u) + \min_{v \in e} X_t(v)| = 2$ with probability $0$;
   - $|\max_{u \in e} X_t(u) + \min_{v \in e} X_t(v)| = 1$ with probability $|x_e^2 - y_e^2|$;
   - $|\max_{u \in e} X_t(u) + \min_{v \in e} X_t(v)| = 0$ with probability $\min(x_e^2, e_e^2)$.

   Assume without loss of generality that $x_e^2 = \min(x_e^2, y_e^2)$. Then, it holds that
   $$\mathbb{E}\left[\left|\max_{u \in e} X_t(u) + \min_{v \in e} X_t(v)\right|\right] = |x_e^2 - y_e^2| = (|x_e| - |y_e|)(|x_e| + |y_e|) = |x_e + y_e| (|x_e| + |y_e|),$$
   where the final equality follows because $x_e$ and $y_e$ have different signs.

These two cases establish (17). Now, we have that
$$\mathbb{E}\left[\sum_{e \in E_H} w(e) \left|\max_{u \in e} X_t(u) + \min_{v \in e} X_t(v)\right|\right] \leq \sum_{e \in E_H} w(e) |x_e + y_e| (|x_e| + |y_e|)$$
$$\leq \sqrt{\sum_{e \in E_H} w(e) |x_e + y_e|^2} \sqrt{\sum_{e \in E_H} w(e)(|x_e| + |y_e|)^2}$$
$$= \sqrt{\sum_{e \in E_H} w(e) \left(\max_{u \in e} f(u) + \min_{v \in e} f(v)\right)^2} \sqrt{\sum_{e \in E_H} w(e)(|x_e| + |y_e|)^2}.$$
By our assumption that $f$ is the eigenvector corresponding to the eigenvalue $\gamma_1$, it holds that
$$\sum_{e \in E_H} w(e) \left(\max_{u \in e} f(u) + \min_{v \in e} f(v)\right)^2 \leq \gamma_1 \sum_{v \in V_H} d(v)f(v)^2.$$
On the other side, we have that
$$\sum_{e \in E_H} w(e)(|x_e| + |y_e|)^2 \leq 2 \sum_{e \in E_H} w(e)(|x_e|^2 + |y_e|^2) \leq 2 \sum_{v \in V_H} d(v)f(v)^2.$$
This gives us that
$$\mathbb{E}\left[\sum_{e \in E_H} w(e) \left|\max_{u \in e} X(u) + \min_{v \in e} X(v)\right|\right] \leq \sqrt{2\gamma_1} \sum_{v \in V_H} d(v)f(v)^2 = \sqrt{2\gamma_1} \cdot \mathbb{E}\left[\sum_{v \in V_H} d(v) |X(v)|\right],$$
which proves the statement. $\qquad\qquad\square$

**Proof of Theorem 1.** Now, we are able to combine these results to prove Theorem 1, which we restate here for completeness.

**Theorem 1** (Main Result). *Given a hypergraph $H = (V_H, E_H, w)$ and parameter $\epsilon > 0$, the following holds:*

1. *There is an algorithm that finds an eigen-pair $(\lambda, f)$ of the operator $J_H$ such that $\lambda \leq \lambda_1(J_G)$, where $G$ is the clique reduction of $H$ and the inequality is strict if $\min_{e \in E_H} r_e > 2$. The algorithm runs in $\mathrm{poly}(|V_H|, |E_H|, 1/\epsilon)$ time.*

2. *Given an eigen-pair $(\lambda, f)$ of the operator $J_H$, there is an algorithm that constructs the two-sided sweep sets defined on $f$, and finds sets $L$ and $R$ such that $\beta_H(L, R) \leq \sqrt{2\lambda}$. The algorithm runs in $\mathrm{poly}(|V_H|, |E_H|)$ time.*

*Proof.* The first statement of the Theorem follows by setting the starting vector $f_0$ of the diffusion to be the minimum eigenvector of the clique graph $G$. By Lemma 7, we have that $R_H(f_0) \leq \lambda_1(D_G^{-1}J_G)$ and the inequality is strict if $\min_{e \in E_H} r_e > 2$. Then, Theorem 3 shows that the diffusion process converges to an eigenvector and that the Rayleigh quotient only decreases during convergence, and so the inequality holds. The algorithm runs in polynomial time since, by Theorem 3, the diffusion process converges in polynomial time, and each step of Algorithm 1 can be computed in polynomial time using a standard algorithm for solving the linear programs.

The second statement is a restatement of Theorem 4. The sweep set algorithm runs in polynomial time since there are $n$ different sweep sets, and computing the hypergraph bipartiteness for each one also takes only polynomial time. $\qquad\square$

## C  Further discussion

### C.1  Discussion of simple reductions

One could naturally ask if we can construct a 2-graph $G$ from the original hypergraph $H$ and apply the 2-graph diffusion process on $G$ to find a good approximation of the cut structure of $H$. However, a simple graph reduction would introduce a factor of $r$, which relates to the rank of hyperedges in $H$, into the approximation guarantee. To see this, we consider the following two natural graph reductions: (1) *Clique Reduction:* From $H = (V, E_H)$, we construct $G = (V, E_G)$ in which every hyperedge $e \in E_H$ of rank $r_e$ is replaced by a clique of $r_e$ vertices in $G$; (2) *Random Reduction:* From $H = (V, E_H)$, we construct $G = (V, E_G)$ in which every hyperedge $e \in E_H$ is replaced by an edge connecting two random vertices of $e$.

To discuss the drawback of the both reductions, we study the following two $r$-uniform hypergraphs $H_1, H_2$, in which all the edges are between the vertex sets $L, R$ and are constructed as follows: (1) in $H_1$, every $e$ that connects $L$ and $R$ contains exactly one vertex from $L$, and $r - 1$ vertices from $R$; (2) in $H_2$, every $e$ that connects $L$ and $R$ contains exactly $r/2$ vertices from $L$ and $r/2$ vertices from $R$. See Figure 6 for illustration. As such, we have that $w_{H_1}(L, R) = |E_{H_1}|$, and $w_{H_2}(L, R) = |E_{H_2}|$.

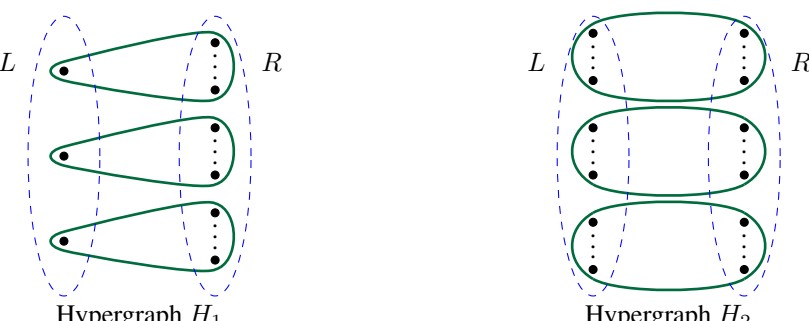

Figure 6: Example hypergraphs illustrating the problem with simple reductions to 2-graphs.

Now we analyse the size of the cut $(L, R)$ in the reduced graph.

- Let $w_G(L, R)$ be the cut value of $(L, R)$ in the *random* graph $G$ constructed from $H$ by the random reduction. We have for $H_1$ that $\mathbb{E}[w_G(L, R)] = \Theta(1/r) \cdot w_{H_1}(L, R)$ and for $H_2$ that $\mathbb{E}[w_G(L, R)] = \Theta(1) \cdot w_{H_2}(L, R)$.
- Similarly, by setting $w_G(L, R)$ to be the cut value of $(L, R)$ in the reduced graph $G$ constructed from $H$ by the clique reduction, we have for $H_1$ that $w_G(L, R) = \Theta(r) \cdot w_{H_1}(L, R)$ and for $H_2$ that $w_G(L, R) = \Theta(r^2) \cdot w_{H_2}(L, R)$.

Since these two approximation ratios differ by a factor of $r$ in the two reductions, no matter how we weight the edges in the reduction, there are always some hypergraphs in which some cuts cannot be approximated better than a factor of $O(r)$. This suggests that reducing a hypergraph $H$ to some 2-graph with some simple reduction would always introduce a factor of $r$ in the approximation guarantee, and that is why a significantly different approach, like what we present in this paper, is needed for hypergraph clustering problems.

## C.2   Computing the minimum eigenvector of $J_H$ is NP-hard

We now show a straightforward but very important fact about the operator $J_H$ and its eigenvalues.

**Theorem 5.** *Given any operator that satisfies a Cheeger-type inequality for hypergraph bipartiteness, there's no polynomial-time algorithm that computes a multiplicative-factor approximation of the minimum eigenvalue or eigenvector unless* $\mathsf{P} = \mathsf{NP}$.

*Proof.* Our proof is based on considering the following MAX SET SPLITTING problem: For any given hypergraph, the problem looks for a partition $L, R$ with $L \cup R = V_H$ and $L \cap R = \emptyset$, such that it holds for any $e \in E_H$ that $e \cap L \neq \emptyset$ and $e \cap R \neq \emptyset$. This problem is known to be NP-complete **?**. This is also referred to as HYPERGRAPH 2-COLORABILITY and we can consider the problem of coloring every vertex in the hypergraph either red or blue such that every edge contains at least one vertex of each color.

We will assume that there is some operator $L$ which satisfies the Cheeger-type inequality given by Lemma 12 and Theorem 4 and that there is an algorithm which can compute the minimum eigen-pair of the operator in polynomial time. We will show that this would allow us to solve the MAX SET SPLITTING problem in polynomial time.

Given a 2-colorable hypergraph $H$ with coloring $(L, R)$, we will use the eigenvector of the operator $J_H$ to find a valid coloring. By definition, we have that $\beta(L, R) = 0$ and $\gamma_1 = 0$ by Lemma 12. Furthermore, by Theorem 4 we can compute disjoint sets $L', R'$ such that $\beta(L', R') = 0$. Note that in general $L'$ and $R'$ will be different from $L$ and $R$.

Now, let $E' = \{e \in E_H : e \cap (L' \cup R') \neq \emptyset\}$. Then, by the definition of bipartiteness, for all $e \in E'$ we have $e \cap L' \neq \emptyset$ and $e \cap R' \neq \emptyset$. Therefore, if we color every vertex in $L'$ blue and every vertex in $R'$ red then every $e \in E'$ will be satisfied. That is, they will contain at least one vertex of each color.

It remains to color the vertices in $V \setminus (L' \cup R')$ such that the edges in $E_H \setminus E'$ are satisfied. The hypergraph $H' = (V \setminus (L' \cup R'), E_H \setminus E')$ must also be 2-colorable and so we can recursively apply our algorithm until every vertex is colored. This algorithm will run in polynomial time since there are at most $O(n)$ iterations and each iteration is polynomial time by our assumption. $\qquad\square$

So, we have answered an open question posed by Yoshida **?** by giving a hypergraph operator which satisfies a Cheeger-type inequality for hypergraph bipartiteness. We have also shown that one cannot hope to compute the minimum eigenvector of *any such operator* unless $\mathsf{P} = \mathsf{NP}$.

## C.3   Discussion about the number of eigenvectors

In this section, we investigate the spectrum of the non-linear hypergraph Laplacian operator introduced in **?** and our new operator $J_H$ by considering some example hypergraphs.

**The hypergraph Laplacian $L_H$ can have more than $2$ eigenvalues.**   Similarly to our new operator $J_H$, for some vector $f$, the operator $L_H$ behaves like the graph Laplacian $L_G = D_G - A_G$ for some graph $G$ constructed by splitting the weight of each hyperedge $e$ between the edges in $S_f(e) \times I_f(e)$. We refer the reader to **?** for the full details of this construction.

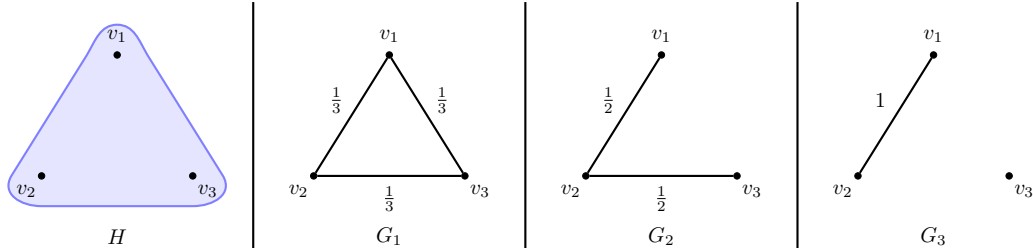

Figure 7: Given the hypergraph $H$, there are three graphs $G_1$, $G_2$, and $G_3$ which correspond to eigenvalues of the hypergraph Laplacian $L_H$.

Now, with reference to Figure 7, we consider the simple hypergraph $H$ where $V_H = \{v_1, v_2, v_3\}$ and $E_H = \{\{v_1, v_2, v_3\}\}$. Letting $f_1 = [1, 1, 1]^\mathsf{T}$, notice that the graph $G_1$ in Figure 7 is the graph constructed such that $L_H$ is equivalent to $L_{G_1}$. In this case,

$$
L_{G_1} = \begin{bmatrix} \frac{2}{3} & -\frac{1}{3} & -\frac{1}{3} \\ -\frac{1}{3} & \frac{2}{3} & -\frac{1}{3} \\ -\frac{1}{3} & -\frac{1}{3} & \frac{2}{3} \end{bmatrix}
$$

and we have

$$
D_H^{-1} L_H f_1 = L_{G_1} f_1 = [0, 0, 0]^\mathsf{T}
$$

which shows that $f_1$ is the trivial eigenvector of $D_H^{-1} L_H$ with eigenvalue 0.

Now, consider $f_2 = [1, -2, 1]^\mathsf{T}$. In this case, $G_2$ shown in Figure 7 is the graph constructed such that $L_H$ is equivalent to $L_{G_2}$. Then, we have

$$
L_{G_2} = \begin{bmatrix} \frac{1}{2} & -\frac{1}{2} & 0 \\ -\frac{1}{2} & 1 & -\frac{1}{2} \\ 0 & -\frac{1}{2} & \frac{1}{2} \end{bmatrix}
$$

and

$$
D_H^{-1} L_H f_2 = L_{G_2} f_2 = \left[ \frac{3}{2}, -3, \frac{3}{2} \right]^\mathsf{T}
$$

and so $f_2$ is an eigenvector of $D_H^{-1} L_H$ with eigenvalue $3/2$.

Finally, we consider $f_3 = [1, -1, 0]$ and notice that the graph $G_3$ is the constructed graph. Then,

$$
L_{G_3} = \begin{bmatrix} \frac{1}{2} & -\frac{1}{2} & 0 \\ -\frac{1}{2} & \frac{1}{2} & 0 \\ 0 & 0 & 0 \end{bmatrix}
$$

and

$$
D_H^{-1} L_H f_3 = L_{G_3} f_3 = [2, -2, 0]^\mathsf{T}
$$

which shows that $f_3$ is an eigenvector of $L_H$ with eigenvalue 2.

Through an exhaustive search through the other possible constructed graphs on $\{v_1, v_2, v_3\}$ we find that these are the only eigenvalues. By the symmetries of $f_2$ and $f_3$ this means that the operator $L_H$ has a total of 7 different eigenvectors and 3 distinct eigenvalues. It is useful to point out that since the $L_H$ operator is non-linear, a linear combination of eigenvectors with the same eigenvalue is *not*, in general, an eigenvector. This example answers an open question in **?** which showed that there are always two eigenvalues and asked whether there can be any more, although further investigation of the spectrum of this operator would be very interesting.

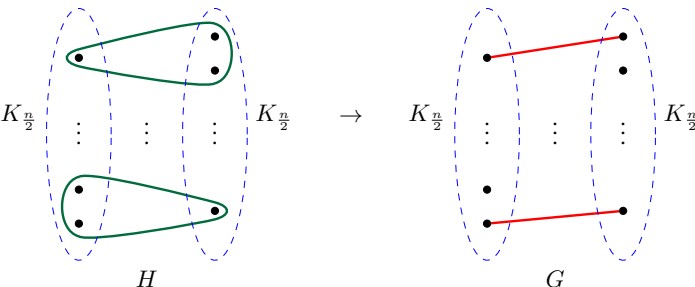

Figure 8: Given the hypergraph $H$, there are $2^{n/3}$ possible graphs $G$, each of which correspond to a different eigenvector of the hypergraph operator $J_H$.

**The hypergraph operator $J_H$ can have an exponential number of eigenvectors.** To study the spectrum of our new operator $J_H$, we construct a hypergraph $H$ in the following way:

- There are $n$ vertices split into two clusters $L$ and $R$ of size $n/2$. There is a clique on each cluster.
- The cliques are joined by $n/3$ edges of rank 3, such that every vertex is a member of exactly one such edge.

Now, we can construct a vector $f$ as follows. For each edge $e$ of rank 3 in the hypergraph, let $u \in e$ be the vertex alone in one of the cliques, and $v, w \in e$ be the two vertices in the other clique. Then, we set $f(u) = 1$ and one of $f(v)$ or $f(w)$ to be $-1$ and the other to be $0$. Notice that there are $2^{n/3}$ such vectors. Each one corresponds to a different graph $G$, as illustrated in Figure 8, which is the graph constructed such that $J_H$ is equivalent to $J_G$ when applied to the vector $f$.

Notice that, by the construction of the graph $H$, half of the edges of rank 3 must have one vertex in $L$ and two vertices in $R$ and half of the rank-3 edges must have two vertices in $L$ and one vertex in $R$. This means that, within each cluster $L$ or $R$, one third of the vertices have $f$-value 1, one third have $f$-value $-1$ and one third have $f$-value 0.

Now, we have that

$$
\begin{aligned}
\left(D_H^{-1} J_H f\right)(u) &= \left(D_H^{-1} J_G f\right)(u) \\
&= \frac{2}{n} \sum_{u \sim_G v} (f(u) + f(v))
\end{aligned}
$$

where $u \sim_G v$ means that $u$ and $v$ are adjacent in the graph $G$. Suppose that $f(u) = 0$, meaning that it does not have an adjacent edge in $G$ from its adjacent rank-3 edge in $H$. In this case,

$$
\begin{aligned}
\left(D_H^{-1} J_H f\right)(u) &= \frac{2}{n} \sum_{u \sim_G v} f(v) \\
&= \frac{2}{n} \left[ \frac{n}{3 \cdot 2} \cdot 1 + \frac{n}{3 \cdot 2} \cdot (-1) \right] \\
&= 0.
\end{aligned}
$$

Now, suppose that $f(u) = 1$, and so it has an adjacent edge in $G$ from its adjacent rank-3 edge in $H$. Then,

$$
\begin{aligned}
\left(D_H^{-1} J_H f\right)(u) &= \frac{2}{n} \sum_{u \sim_G v} (1 + f(v)) \\
&= \frac{2}{n} \left[ \frac{n}{2} + \frac{n}{3 \cdot 2} \cdot (-1) + \left( \frac{n}{3 \cdot 2} - 1 \right) \cdot 1 - 1 \right] \\
&= 1 - \frac{4}{n}.
\end{aligned}
$$

Similarly, if $f(u) = -1$, we find that

$$
\left(D_H^{-1} J_H f\right)(u) = \frac{4}{n} - 1
$$

and so we can conclude that $f$ is an eigenvector of the operator $J_H$ with eigenvalue $(n-4)/n$. Since there are $2^{n/3}$ possible such vectors $f$, there are an exponential number of eigenvectors with this eigenvalue. Once again, we highlight that due to the non-linearity of $J_H$, a linear combination of these eigenvectors is in general not an eigenvector of the operator.