# OpenReview forum: "Finding Bipartite Components in Hypergraphs"
_NeurIPS.cc/2021/Conference — NeurIPS 2021 Poster_

### Official Review · Reviewer_PH5A · 2021-07-15

**Rating:** 7
**Confidence:** 4

**Summary:**

This paper proposes a polynomial-time algorithm that finds densely connected bipartite components in a hypergraph. The algorithm is based on a heat diffusion process generalized from graphs to hypergraphs. Cheeger-type approximation guarantee is established. Empirical results show that the new method has superior performance for both synthetic and real hypergraphs.

**Limitations And Societal Impact:**

I cannot think of any potential negative societal impact based on this work.

**Main Review:**

Hypergraphs and related nonlinear Laplacian operators received renewed interests recently in both machine learning and applied mathematics communities. The heat diffusion process studied in this paper is a nice addition, in particular it applies to find bipartite components in hypergraphs, which is a different application from most previous works. At the same time, the nonlinear operator considered in this work is simple and intuitive.

Technical analyses is this paper appear to be clear and solid. Overall I think the paper is well written and it solves an interesting problem. Empirical results are promising and further support the usefulness of the proposed method.

I have a few minor comments:

- line 129-131: Why is it intuitive that the rate of change should involve only the maximum and minimum values? Apparently this is the simplest case, but in more complex settings (e.g. when one models more complex higher-order relations as in [22]) can it depend on other node values as well? The formulation considered in this paper seems to correspond to hypergraphs modelled by all-or-nothing cut (or equivalently the unit cut-cost in [13]). Can the heat diffusion process be generalized to more complex higher-order relations among nodes within a hypergraph? More explanations are needed.

- Baseline: Following footnote 4, another baseline could be $f_1(\mathcal{J}_G)$. Although the authors explained in the supplementary material that the worst-case performance guarantee using $f_1(\mathcal{J}_G)$ is scaled by a factor of $r$ the rank of hyperedge, it would provide additional information to the reader about the practical importance of the new method.

- Synthetic experiments: Since you have the ground-truth information can you also show the F1 scores?

**Time Spent Reviewing:**

4

---

> ### Author Response · Authors · 2021-08-09
> **Response to Reviewer PH5A**
>
> Thank you for your detailed and helpful review. We have addressed your comments below and will be happy to continue the discussion if you have any remaining questions.
>
> ### Why does the diffusion involve only the maximum and minimum values?
> This is an excellent question and it warrants some discussion, particularly with respect to the previous work on submodular hypergraphs which you mention [22, 35]. We include several remarks below which address your point from a variety of perspectives, and we hope that this answers your question.
>
> Firstly, we feel it is a reasonable high-level intuition that in a diffusion process on a hypergraph, the extreme values within a given edge should be the most important when determining how the diffusion progresses. Therefore, as you say, we consider the simplest case in which the diffusion happens on the extreme vertices only.
>
> Additionally, one can view our new operator $J_H$ as an analogue of the hypergraph Laplacian operator $L_H$ proposed in [5]. Roughly speaking, $L_H$ extends the graph Laplacian to hypergraphs, and our $J_H$ extends the *signless Laplacian*. As such, we are following [5] in which only the extreme vertices within an edge contribute to the diffusion process. One consequence is that the quadratic form of our operator $J_H$ has the same form as the quadratic form of $L_H$, where $(\max_{u \in e} f_t(u) - \min_{v \in e} f_t(v))$ is replaced with $(\max_{u \in e} f_t(u) + \min_{v \in e} f_t(v))$. This is analogous to the difference between the quadratic forms of the graph Laplacian and signless graph Laplacian.
>
> At a more technical level, our algorithm uses the two-sided sweep set method proposed by Trevisan [29] to find the sets L and R. In this algorithm, we find some threshold $t$ and all vertices with f-value less than $-t$ are included in L and all vertices with f-value greater than $t$ are included in R (see Lines 169-172 in the submission). As such, it is only the values of the maximum and minimum vertices within an edge which determine whether it will be cut by L and R and so it is reasonable to design the diffusion process to balance only the extreme values within each edge.
>
> Finally, as you correctly point out, we consider the "all-or-nothing cut" although previous work in spectral hypergraph theory has generalised the hypergraph Laplacian from [5] to *submodular hypergraphs* in which each edge is associated with an arbitrary submodular cut function [22, 35]. We have no reason to believe that our approach could not also be generalised in this way. We will add a comment in the final version to point this out, and leave it as an open question for future work.
>
> ### Baseline algorithm
> The baseline algorithm we compare with is Trevisan's sweep set algorithm [29] applied to the clique reduction of the hypergraph. I.e. we apply the two-sided sweep set to $f_1(\mathcal{J}_G)$ where $G$ is the clique reduction of the hypergraph.
> This appears to be what you are suggesting but if we have misunderstood you or anything remains unclear we would be happy to discuss it.
> In our experiments we find that our new algorithm significantly out-performs this baseline at a cost of only a constant factor in the runtime, which demonstrates the practical importance of our approach.
>
> ### F1-scores for synthetic experiments
> Thank you for your suggestion to report the F1 scores for the experiments on synthetic datasets. We will extend Figure 4 in the final version with a plot of the F1 scores. We have given a table of results below.
>
> |  n   | r |   p   | q/p | FBCA F1-Score | CC F1-Score |
> |------|---|-------|-----|---------------|-------------|
> | 2000 | 5 | 1e-11 | 15  |   **0.774**   |    0.764    |
> | 2000 | 5 | 1e-11 | 20  |   **0.917**   |    0.871    |
> | 2000 | 5 | 1e-11 | 25  |   **0.980**   |    0.919    |
> | 2000 | 5 | 1e-11 | 30  |   **0.995**   |    0.953    |

---

### Official Review · Reviewer_Xs5P · 2021-07-16

**Rating:** 6
**Confidence:** 3

**Summary:**

This paper proposes a diffusion-based algorithm for finding bipartite components in hypergraphs. The work rigorously analyzes the algorithm and provides theoretical guarantees.

**Main Review:**

This paper uses a diffusion-based algorithm that requires multiple iterations for convergence.  Therefore, it comes with high computational complexity. The complexity analysis of algorithm is missing. Further, the empirical time complexity (Table 1) shows that the algorithm is not scalable to large-scale networks with millions of nodes where the problem is much more relevant. More experimental results and computation complexity justification is required to assess the practical significance of the work. The work extends the theoretical understanding of the diffusion process, especially for hyper-graphs which is a major strength of this work.



**Time Spent Reviewing:**

5

---

> ### Author Response · Authors · 2021-08-06
> **Response to Reviewer Xs5P**
>
> Many thanks for your time and helpful review of our work. We have addressed each of your specific points below. Please let us know if you have any further questions and we will be happy to discuss them.
>
> ### Complexity analysis of the algorithm
> Since the goal of this paper is not to find the fastest possible algorithm for the problem, we have not given the exact computational complexity of our new algorithm. As we mention on Line 71 of the submission, the exact complexity bound would depend on the choice of linear program solver and we do not optimize this in our implementation.
>
> However, we do prove in Theorem 1 that the time complexity of our algorithm is polynomial in the size of the input. Moreover, as an intermediate step in the proof, we prove Theorem 3 (i.e., Line 226 in the Appendix and the paragraph afterwards) which gives an explicit polynomial bound on the number of iterations of the algorithm before convergence.
>
> In the final version, we will highlight the results which relate to the running time of the algorithm and add a comment giving an exact bound, making some assumption about the complexity of the linear program solver.
>
> ### Scalability of the algorithm on large networks
> We highlight that we have run experiments on the Penn-Treebank dataset which contains roughly 1 million vertices (see Lines 296 – 303 of the submission), and we find that the algorithm runs in reasonable time (~2 minutes on a standard desktop machine); in our point of view, experiments on graphs of such scale should be sufficient for a research paper, especially given that the main contribution of our work is towards developing some theory (as you insightfully pointed out as a major strength of our work). We also remark that, for many applications, clustering algorithms will be used as a one-off data analysis step and therefore a running time in the order of a few minutes may be sufficient.
>
> We have shown experimentally that our algorithm *does* scale to graphs with roughly 1 million vertices, and we believe that this is a good result given that ours is the first algorithm designed to solve this problem. We agree that, similar to many spectral graph algorithms, our algorithm may not scale for some applications on datasets of a larger scale, and that developing more scalable spectral hypergraph algorithms is a very interesting area for future research.
>
> ### Additional experimental results
> We are unsure what kind of additional experimental results you feel would be useful. However, if you have any specific suggestions on the setup of the experiments, datasets on which our algorithm should run, or baseline algorithms against which our algorithm should compare, we would be happy to follow your suggestions and add more experimental results during the later discussion period or in the final version of our paper.

---

### Official Review · Reviewer_SWke · 2021-07-17

**Rating:** 6
**Confidence:** 3

**Summary:**

The paper discusses how to find bipartite components in a hyper graph.  A bipartite component consists of two subsets of vertices L and R.  Bipartite implies  there is minimal connection within L as well as within R.  Component implies that there is minimal connection between vertices in L union R and vertices outside.

The algorithm is based on diffusion process to find sets with small hyper graph bipartiteness.  The paper provides a couple of theorems for the existence of a polynomial algorithm.  Experiments are conducted on both synthetic as well as real-world datasets, including Penn Treebank and DBLP.

**Limitations And Societal Impact:**

The paper itself does not contain an ethics statement, though the authors acknowledged that they have read the ethics guidelines.

As suggestions for improvements, the paper would be a lot stronger if it could be motivated by real applications that someone actually wants to solve.  As of now the evaluations are rather contrived.

There should also be some discussions on the large increase of computational time relative to CC, and what could be done about it or why the increase is justified.

**Main Review:**

On the plus side, the paper gives a formal treatment of the problem.  It is written rigorously, if rather densely, but the key ideas come across.  It is also good that the experiments explore both synthetic datasets (to look at the runtime with increasing size) as well as real-world datasets (to explore potential applications).

One weakness of the paper is that it is light on applications.  It is not clear to me whether there will be many occasions in which we'd be looking for bipartite components in a hyper graph.  While the paper gives an example on Peen Treebank, claiming that it is achieving good performance given that it is an unsupervised general purpose algorithm, this seems to still be a contrived example.  The discussion on DBLP is similarly vague, saying that the algorithm can separate authors from conferences, but I imagine that task wouldn't really be an actual application.

Another point is the computational complexity relative to the baseline clique cut algorithm, which is much faster (by an order of magnitude or thereabout).  While the proposed method produces "some" improvement, some discussion on whether the difference justifies the large increase in computational cost.

**Time Spent Reviewing:**

1

---

> ### Author Response · Authors · 2021-08-06
> **Response to Reviewer SWke**
>
> Many thanks for your insightful review of our paper. We are pleased that the key ideas in the paper are clear and we have addressed each of your specific points below. Please let us know if you have any further questions and we will be happy to discuss them.
>
> ### Applications of the algorithm
> Thank you for your comment about the applications of our work. We acknowledge that the experiments in our paper could be seen as a "proof of concept" although we also highlight the following points.
>
> One can view our algorithm as a clustering method for *disassortative* hypergraphs, meaning that we expect edges to contain vertices with different class labels. Clustering disassortative *graphs* is an area of active research in the machine learning community [20, 24, 38], and it is natural to consider the generalisation of this problem to hypergraphs. Since we are among the first to consider this generalisation, the availability of disassortative hypergraph datasets is quite limited for now.
>
> Furthermore, we consider the most significant contribution of this paper to be our theoretical contribution to the spectral theory of hypergraphs. This new spectral hypergraph theory is a relatively recent development and as pointed out by Reviewer PH5A, it is of growing interest in both the machine learning and applied mathematics communities. We hope that our work could inspire further research to generalise results from spectral graph theory to hypergraphs using non-linear Laplacian-type operators.
>
> Our experiments give further evidence that the spectral theory of hypergraphs can be used to find meaningful structure in real-world data, complementing the experimental work in e.g. [22, 28]. We expect there will be further experimental and applied work in this area in the future and that spectral hypergraph theory will become an important tool for the analysis of hypergraphs.
>
> ### Runtime comparison with CliqueCut
> On Line 286 of the submission we mention that in practice the difference between our algorithm's runtime and the runtime of CliqueCut appears to be a constant factor. This means that (i) the scalability of our algorithm to large graphs will be similar to that of the baseline algorithm and (ii) we expect that our new algorithm will be chosen for one-off clustering tasks since it is worth a constant factor in order to achieve better performance.
>
> Finally, as we mention on Line 71 of the submission, the goal of this paper is not to find the fastest possible algorithm and we agree that it would be a very interesting research direction to explore faster spectral hypergraph algorithms. We will add a comment to this effect in the final version.
>
> ### Ethics Statement
> We did not include an ethics statement in the main document because (i) we felt there were no privacy concerns since our new unsupervised algorithm does not require training data and we apply it only to widely available datasets, and (ii) we cannot see any possible negative societal impact from our theoretical contributions.
>
> We will be happy to add an ethics statement in the final version if you think that it is needed.

---

### Decision · Program_Chairs · 2021-09-27

**Decision:**

Accept (Poster)

**Comment:**

The paper provides a clean solution for a new problem, but the motivations and applications of the paper is not completely clear.  The results is not straightforward. The applications and motivating examples presented in the paper, on the other hand, are not  convincing.  Given the results are mathematically interesting and the area is of interest to the community, the paper could be accepted as a poster.